# Multi-LLM Adaptive Conformal Inference for Reliable LLM Responses

**Kangjun Noh**[1], **Seongchan Lee**[2], **Ilmun Kim**[2], **Kyungwoo Song**[1]*

[1]Department of Applied Statistics and Data Science, Yonsei University
[2]Department of Mathematical Sciences, KAIST

## Abstract

Ensuring factuality is essential for the safe use of Large Language Models (LLMs) in high-stakes domains such as medicine and law. Conformal inference provides distribution-free guarantees, but existing approaches are either overly conservative, discarding many true-claims, or rely on adaptive error rates and simple linear models that fail to capture complex group structures. To address these challenges, we reformulate conformal inference in a multiplicative filtering setting, modeling factuality as a product of claim-level scores. Our method, Multi-LLM Adaptive Conformal Inference (**MACI**), leverages ensembles to produce more accurate factuality-scores, which in our experiments led to higher retention, while validity is preserved through group-conditional calibration. Experiments show that MACI consistently achieves user-specified coverage with substantially higher retention and lower time cost than baselines. Our repository is available at https://github.com/MLAI-Yonsei/MACI.git.

## 1 Introduction

As the performance of Large Language Models (LLMs) continues to advance, attempts to directly utilize their responses in high-stakes domains such as medicine and law are increasing. However, studies continue to report that LLM responses may contain false information (Wang et al., 2024). Therefore, to use LLMs reliably in these critical fields, guaranteeing the factuality of their responses has emerged as an important challenge.

Various methods have been proposed to guarantee the factuality of LLMs, but some are difficult to apply to black-box models (Meng et al., 2022; Zhang et al., 2025; Chen et al., 2024a) or require access to large external databases or online databases (Chen et al., 2024b; Lee and Yu, 2025). Sampling-based methods (Manakul et al., 2023; Sawczyn et al., 2025) are relatively free from the constraints, but the process of repeatedly checking for response consistency incurs considerable time and financial costs, and they face difficulties in rigorously providing a statistical guarantee at a user-specified error rate.

Recently, studies applying Conformal Inference (CI) (Papadopoulos et al., 2002; Vovk et al., 2005; Lei et al., 2018; Angelopoulos and Bates, 2022) to guarantee the factuality of LLMs have been proposed. For instance, Mohri and Hashimoto (2024) apply the concept of CI to the existing framework of decomposing LLM responses into independent claims and assigning a factuality-score to each one. Their method proposes filtering out claims that do not pass a predetermined threshold. However, because this single, global threshold is applied uniformly to all data, it provides only marginal coverage and can be overly conservative, resulting in the removal of a lot of true information. To improve upon this, Cherian et al. (2024) introduce conditional conformal inference. Instead of a single static threshold for all data, this method employs a threshold function that allows it to change based on the characteristics of a given sample. But it relies on adaptive error rates that are unsuitable for high-stakes applications requiring a fixed guarantee. Its threshold function also struggles to capture the characteristics of the complex grouping criteria of LLM responses that are separated by their semantic properties. Fundamentally, they typically define conformity score

---

*Corresponding Author.

**BCI**
(Mohri and Hashimoto et al.)

~~Amitriptyline and alprazolam are both medications,~~ [T] ~~but they are used to treat different conditions~~ [T] ~~and have different potential side effects~~ [T]. Amitriptyline and alprazolam have different mechanisms of action. [T] ~~Amitriptyline is a tricyclic antidepressant~~ [T] ~~used to treat depression~~ [T] ~~and certain types of chronic pain,~~ [T] ~~but it is not used for anxiety disorders.~~ [F] ~~Alprazolam, on the other hand, is a benzodiazepine medication~~ [T] ~~primarily used to treat anxiety disorders~~ [T] ~~and panic attacks.~~ [T] ~~It works by enhancing the effects of a neurotransmitter in the brain~~ [T] ~~called GABA,~~ [T] ~~which helps to reduce anxiety~~ [T] ~~and promote relaxation.~~ [T] ~~Both medications can cause side effects~~ [T] ~~including drowsiness~~ [T] ~~and dizziness,~~ [T] ~~though confusion is not a typical side effect.~~ [F] ~~However, alprazolam is more likely to cause dependence~~ [T] ~~and withdrawal symptoms,~~ [T] ...

Target Coverage: High

Retention: Low

**CCI**
(Cherian et al.)

Amitriptyline and alprazolam are both medications, [T] but they are used to treat different conditions [T] and have different potential side effects. [T] Amitriptyline and alprazolam have different mechanisms of action. [T] Amitriptyline is a tricyclic antidepressant [T] used to treat depression [T] and certain types of chronic pain, [T] but it is not used for anxiety disorders. [F] Alprazolam, on the other hand, is a benzodiazepine medication [T] primarily used to treat anxiety disorders [T] and panic attacks. [T] It works by enhancing the effects of a neurotransmitter in the brain [T] called GABA, [T] which helps to reduce anxiety [T] and promote relaxation. [T] Both medications can cause side effects [T] including drowsiness [T] and dizziness, [T] though confusion is not a typical side effect. [F] However, alprazolam is more likely to cause dependence [T] and withdrawal symptoms, [T] ...

Target Coverage: Not Enough

Retention: High

**MACI**
(Ours)

Amitriptyline and alprazolam are both medications, [T] but they are used to treat different conditions [T] and have different potential side effects. [T] Amitriptyline and alprazolam have different mechanisms of action. [T] Amitriptyline is a tricyclic antidepressant [T] used to treat depression [T] and certain types of chronic pain, [T] but it is not used for anxiety disorders. [F] Alprazolam, on the other hand, is a benzodiazepine medication [T] primarily used to treat anxiety disorders [T] and panic attacks. [T] It works by enhancing the effects of a neurotransmitter in the brain [T] called GABA, [T] which helps to reduce anxiety [T] and promote relaxation. [T] Both medications can cause side effects [T] including drowsiness [T] and dizziness, [T] ~~though confusion is not a typical side effect.~~ [F] However, alprazolam is more likely to cause dependence [T] and withdrawal symptoms, [T] ...

Target Coverage: High

Retention: High

Figure 1: Comparison of Conformal Inference Methods. T (true) and F (false) denote ground-truth labels per claim. Basic Conformal Inference (Mohri and Hashimoto, 2024) attains coverage by aggressive filtering, yielding low retention. Conditional Conformal Inference (Cherian et al., 2024) proposes adaptive thresholds but relaxes guarantees; MACI achieves both high coverage and retention.

based on a single worst-case score (e.g., the highest confidence score among false-claims), which ignores the collective confidence of other claims and renders the calibration process highly sensitive to estimation error of worst-case score, leading to filtering many true-claims out.

In this context, we propose a new methodology called Multi-LLM Adaptive Conformal Inference (MACI). The core objective of MACI is to preserve as much factual information as possible while strictly adhering to a low user-specified error rate. Assuming an ideal oracle factuality-score, we derive an explicit filtering rule which is written as a cumulative product of probabilities that retains the maximum number of claims subject to a target coverage constraint. Inspired by this finding, we design a new adaptive CI framework that uses a conformity score in the form of a cumulative product of estimated factuality-scores. In contrast to previous methods that calculate conformity scores relying on the single worst-case score, the conformity score of MACI aggregates estimated factuality-scores between claims, significantly improving the robustness of the calibration. We further theoretically prove that the quality of the factuality-score directly impacts the efficiency of MACI, which is quantified by retention ratio. Accordingly, we adapt our strategy to maximize factuality-score quality through a multi-LLM ensemble. As a result, MACI not only theoretically guarantees group-conditional coverage but empirically demonstrates robust group-conditional coverage across diverse datasets, all while showing a substantially higher retention ratio than existing methodologies. Figure 1 shows an actual example demonstrating MACI's superior coverage and retention ratio compared to existing conformal inference methods.

Our main contributions are:

- We introduce a multiplicative filtering framework that models factuality as the product of claim-level scores (factuality-score) while preserving finite-sample guarantees.
- We provide, to our knowledge, the first retention theoretical analysis in conformal inference, linking oracle–estimator deviations to true-claim preservation and motivating ensemble design.
- We extend conformal inference with group-conditional calibration and a multi-LLM ensemble, ensuring group-conditional coverage and showing substantially higher retention than conformal baselines in high-stakes domains.

## 2 RELATED WORKS

Existing Basic Conformal Inference (BCI) (Mohri and Hashimoto, 2024) performs false-claim filtering by applying a factuality-score-based threshold to individual claims within LLM responses, providing a distribution-free guarantee of an error rate below $\alpha$ over the entire data. However, because BCI provides this guarantee only over the entire sample distribution (marginal coverage), specific subgroups may experience group-specific under- or over-coverage, and biases may persist.

Much research in the CI field has been conducted to move beyond this marginal coverage and provide group-conditional coverage guarantees (Vovk, 2012). Hébert-Johnson et al. (2018) propose

enhanced predictive guarantees that simultaneously satisfy calibration across various computationally definable subgroups through multicalibration. Jung et al. (2023) propose the batch multivalid conformal prediction that introduces the concept of multivalid coverage where coverage is guaranteed simultaneously across multiple groups and various threshold levels. This approach further strengthens simple group-conditional coverage. In practice, when such guarantees are applied, while coverage deviation across groups decreases, the prediction set tends to become larger and retention tends to be lower. Ding et al. (2023) and Gao et al. (2025) analyze the trade-off between group-based coverage and efficiency (set size) by clustering classes or groups or by combining surrogate information, and they propose improvements. Liu and Wu (2025) apply these group-conditional coverage methodologies to ensure the factuality of LLM responses. They systematically evaluate factuality guarantee performance across demographic subgroups by applying multicalibration and multivalid CP to claim-level score calibration and document-level conformal prediction. Detommaso et al. (2024) enhance the reliability of LLM confidence itself through group-specific multicalibration using embeddings and self-annotation. However, stronger group-conditional or multivalid guarantees typically require more conservative prediction sets, which can hurt efficiency (e.g., larger sets or more abstentions) in standard conformal settings. When such methods are naively applied to LLM response filtering, this conservatism can translate into lower retention, raising concerns about practical usability in high-throughput applications. Our research continues the context of guaranteeing group-conditional coverage, but focuses on maintaining a practically high retention ratio while ensuring group-conditional coverage under realistic group definitions based on specific features. Feng et al. (2025) propose a framework that ensures group-conditional coverage while increasing retained claims by applying RAG (Lewis et al., 2021) to the calibration and filtering. However, this approach essentially shifts the application of conformal inference from the LLM to external components like the embedding model and retriever.

Cherian et al. (2024) share the same goal with MACI, which is retaining as many true-claims as possible under group-conditional coverage. They extend conditional conformal methods (Gibbs et al., 2025) to learn sample-specific thresholds from calibration data. They propose a framework that achieves conditional guarantees based on groups defined by prompt and response features while actively improving true-claim retention through techniques like adaptive error rate and conditional boosting. This approach of simultaneously pursuing the two practical goals of group-conditional coverage and retention ratio presents a new balance point between ensuring factuality in LLM responses per group and preserving information. Meanwhile, limitations such as the practicality of adaptive $\alpha$ in high-risk environments and the limited expressiveness of the threshold function still remain.

## 3 BACKGROUND AND PRELIMINARIES

**Document structure and factuality-scores.** Let $\mathcal{P}$ denote the space of prompts and $\mathcal{C}$ the space of claims. Each document $D = (P, C, Y)$ consists of a prompt $P \in \mathcal{P}$, a set of claims $C = \{c_1, \ldots, c_{|C|}\} \subseteq \mathcal{C}$, and labels $Y \in \{0, 1\}^{|C|}$ indicating which claims are factual (true-claims). We assume documents are drawn i.i.d. from a distribution $\mathbf{P}$, which implies exchangeability of calibration and test data. A factuality-score function $p : \mathcal{P} \times \mathcal{C} \to [0, 1]$ assigns each $(P, c)$ the probability of being factual, with oracle $p^*$ and estimator $\hat{p}$.

**Filtering operator.** Given a score function $p$, a threshold $\tau \in [0, 1]$, and optional randomization $U$, the filtering operator $F(p, \tau, U; P, C) \subseteq C$ returns the claims retained under $\tau$. We denote by $F_{n,\alpha}$ the data-driven version of this operator, obtained by calibrating $\tau$ on held-out data to achieve target level $\alpha$; this calibrated rule is the central object of our analysis.

**Group-conditional coverage.** Exact document-level coverage is infeasible in a distribution-free setting (Vovk, 2012; Foygel Barber et al., 2021). Instead, we require validity within subgroups that capture meaningful distinctions such as domains, topics, or user populations. The held-out example is a full document $D_{n+1} = (P_{n+1}, C_{n+1}, Y_{n+1})$, drawn i.i.d. from the same distribution as the calibration data. Formally, let $g : \mathcal{P} \times \mathcal{C} \to \{1, \ldots, K\}$ be a grouping function that assigns each document $(P_i, C_i)$ to one of $K$ groups. Each document consists of a prompt $P_i$ and a set of generated claims $C_i = \{c_{i,1}, \ldots, c_{i,N_i}\}$, where $N_i = |C_i|$ denotes its number of claims. Let $A_i = \{c_{i,j} \in C_i : y_{i,j} = 1\}$ denote the true-claim set of document $D_i$. We then require that, for every group $k \in \{1, \ldots, K\}$,

$$\mathbb{P}(F_{n,\alpha}(P_{n+1}, C_{n+1}) \subseteq A_{n+1} \mid g(P_{n+1}, C_{n+1}) = k) \geq 1 - \alpha. \tag{1}$$

This mirrors the Mondrian conformal prediction framework (Vovk et al., 2005), but is applied here to prompt–claim pairs rather than individual data points. In our experiments, we instantiate $g$ using high-level, dataset-specific categories (e.g., medical question types or entity groups).

# 4  MACI: MULTI-LLM ADAPTIVE CONFORMAL INFERENCE

Building on Section 3, our goal is to design a filtering rule $F_{n,\alpha}$ that satisfies the group-conditional coverage guarantee (1). The baseline BCI method applies one global threshold, which is simple but ignores group heterogeneity and relies on a single predictor. Inspired by adaptive conformal prediction (Romano et al., 2020), we propose MACI, which aggregates scores from multiple LLMs and calibrates group-conditional thresholds. This approach improves retention while maintaining the required coverage guarantees.

## 4.1  ORACLE FACTUALITY

Given the definition of a factuality-score function in Section 3, we first consider an idealized regime where the factuality-score coincides with the true conditional probability. In this oracle setting, the model has complete distributional knowledge of claim correctness, so that for any prompt–claim pair $(P, c)$ it can evaluate $\mathbb{P}(y = 1 \mid P, c)$ exactly.

**Definition 1** (Oracle Filtering Rule)**.** For any prompt-claim pair $(P, c)$ with binary factuality label $y \in \{0, 1\}$, the oracle factuality-score is defined as $p^*(P, c) := \mathbb{P}(y = 1 \mid P, c)$. For a document $D_i = (P_i, C_i, Y_i)$ with claim set $C_i = \{c_{i,1}, \ldots, c_{i,N_i}\}$, the oracle score for each claim is

$$p_i^*(c_{i,j}) := p^*(P_i, c_{i,j}) = \mathbb{P}(Y_{i,j} = 1 \mid P_i, c_{i,j}).$$

Let $[N_i] = \{1, 2, \ldots, N_i\}$ and $\pi_i : [N_i] \to [N_i]$ be a permutation that orders the claims by decreasing oracle scores, $p_i^*(c_{i,\pi_i(1)}) \geq \cdots \geq p_i^*(c_{i,\pi_i(N_i)})$, with ties broken arbitrarily. Define $\prod_k := \prod_{j=1}^k p_i^*(c_{i,\pi_i(j)})$ with the convention $\prod_0 = 1$ and $\prod_{N_i+1} = 0$. For a threshold $\tau \in [0, 1]$, define the cutoff index and filtered set

$$K_i^*(\tau) := \max \left\{ k \in [N_i] : \prod_{j=1}^k p_i^*(c_{i,\pi_i(j)}) \geq \tau \right\}, \quad F_\tau^*(P_i, C_i) := \{c_{i,\pi_i(j)} : j \leq K_i^*(\tau)\}.$$

with the convention $\max \emptyset = 0$. Thus $F_\tau^*$ ensures coverage at level $\tau$ and is monotone in $\tau$ ($\tau_1 \leq \tau_2 \Rightarrow F_{\tau_2}^* \subseteq F_{\tau_1}^*$), But it is conservative since coverage typically exceeds $\tau$. To obtain *exact* coverage, we randomize at the boundary index $K_i^*(\tau)$. Define $\gamma_i(\tau) = \frac{\Pi_{K_i^*(\tau)} - \tau}{\Pi_{K_i^*(\tau)} - \Pi_{K_i^*(\tau)+1}} \in [0, 1]$ (with $\gamma_i(\tau) = 0$ if the denominator vanishes). With $U_i \sim \mathrm{Unif}(0, 1)$, the randomized oracle rule is

$$F(p^*, \tau, U_i; P_i, C_i) = \begin{cases} \{c_{i,\pi_i(j)} : j \leq K_i^*(\tau)\}, & U_i > \gamma_i(\tau), \\ \{c_{i,\pi_i(j)} : j \leq K_i^*(\tau) + 1\}, & U_i \leq \gamma_i(\tau). \end{cases}$$

This randomization balances inclusion and exclusion at the boundary, achieving exact coverage at level $\tau$ while maximizing expected retention ratio.

## 4.2  ADAPTIVE CONFORMAL INFERENCE FOR FALSE-CLAIM FILTERING

Before presenting our adaptive conformal inference (ACI) framework, we clarify the perspective taken in this section. Conformal inference aims to guarantee validity by designing a filtering rule that achieves a target coverage level. In principle, one could satisfy this requirement by discarding most claims, but such a strategy would be overly conservative. Thus, after ensuring validity, we examine how efficiency is affected when using an estimated factuality score rather than the oracle score, and how this behavior compares to the oracle benchmark under simple idealized assumptions.

Based on this distinction, the remainder of this section first explains how our method secures **validity**, and then analyzes its **efficiency** in producing informative filtering sets.

**Validity.**  As discussed in Section 4.1, the oracle filtering rule requires access to the oracle factuality-score $p^*$ and therefore serves only as a theoretical benchmark. In practice, $p^*$ is unknown and should be replaced with an estimated score $\hat{p}$ obtained from a black-box verifier (e.g., an LLM). The resulting filtering operator $F(\hat{p}, \tau, U; P, C)$ inherits the oracle structure but depends critically on how the threshold $\tau$ is chosen. The objective is to calibrate $\tau$ in a data-driven manner so that, with high probability, all retained claims are factual, thereby achieving valid finite-sample coverage. We estimate $\tau$ using conformal quantiles computed on held-out calibration data. This calibration step guarantees coverage even when $\hat{p}$ is only an imperfect approximation of the oracle score $p^*$.

To carry out this calibration, we require a conformity score that captures the document-level filtering event in a scalar form. Recall that for each document $(P_i, C_i, Y_i)$, $A_i = \{c_{i,j} \in C_i : y_{i,j} = 1\}$ denote the set of true-claims, and let $U_i \sim \text{Unif}(0, 1)$ be the randomization variable. We define the conformity score $E_i = \inf\{\tau \in [0, 1] : F(\hat{p}, \tau, U_i; P_i, C_i) \subseteq A_i\}$, which represents the smallest threshold at which all retained claims are true-claims. Each $E_i$ compresses the document-level filtering requirement into a single scalar quantity, making it directly suitable for conformal quantile calibration (Lemma 1 in Appendix A). Applying the standard conformal quantile argument then yields the following finite-sample guarantee.

**Theorem 1** (Marginal Coverage Guarantee). *If the samples $(P_i, C_i, Y_i)$, for $i \in \{1, \ldots, n+1\}$, are exchangeable, ACI (Algorithm 1) satisfies*

$$\mathbb{P}\big(F_{n,\alpha}(P_{n+1}, C_{n+1}) \subseteq A_{n+1}\big) \geq 1 - \alpha$$

*Furthermore, if the scores $E_i$ are almost surely distinct, the marginal coverage is nearly tight:*

$$\mathbb{P}\big(F_{n,\alpha}(P_{n+1}, C_{n+1}) \subseteq A_{n+1}\big) \leq 1 - \alpha + \frac{1}{n+1}.$$

Marginal validity ensures that the overall error rate is controlled on average, but this may hide differences between groups, with some subpopulations receiving weaker guarantees. To address this, we extend adaptive conformal inference to a group-conditional setting, so that validity is enforced separately for each group. This extension follows the Mondrian conformal framework of Vovk et al. (2005). Instead of pooling all scores, calibration is restricted to examples from the same group as the test instance. For group $k$ with calibration set $\mathcal{I}_k = \{i : g(P_i, C_i) = k\}$, the threshold is $\hat{Q}_{1-\alpha}^{(k)} = \text{Quantile}(\{E_i : i \in \mathcal{I}_k\}, 1 - \alpha)$. Given a test instance in group $k$, the filter is $F_{n,\alpha}^{(k)}(P_{n+1}, C_{n+1}) = F(\hat{p}, \hat{Q}_{1-\alpha}^{(k)}, U_{n+1}; P_{n+1}, C_{n+1})$.

**Theorem 2** (Group-conditional Coverage Guarantee). *If the samples $\{(P_i, C_i, Y_i)\}_{i=1}^{n+1}$ are exchangeable, the group-conditional conformal inference rule satisfies*

$$\mathbb{P}\big(F_{n,\alpha}^{(k)}(P_{n+1}, C_{n+1}) \subseteq A_{n+1} \,|\, g(P_{n+1}, C_{n+1}) = k\big) \geq 1 - \alpha,$$

*for all $k \in \{1, \ldots, K\}$ with $\mathbb{P}(g(P_{n+1}, C_{n+1}) = k) > 0$.*

A key implication of Theorem 2 is that it ensures *finite-sample, distribution-free validity within each group*, in contrast to the marginal guarantee of Theorem 1, which holds only in aggregate. Each group achieves a level $1 - \alpha$ based on its own calibration size $n_k$, ensuring that even small groups are covered, albeit with more conservative thresholds and reduced retention.

**Efficiency.**  Group-conditional validity holds for any estimated score $\hat{p}$, but validity alone does not determine how many claims are retained. A filtering rule may satisfy the coverage requirement while admitting only a small set of claims, yielding outcomes that are valid yet not practically useful. This motivates analyzing efficiency alongside validity.

To characterize the efficiency attainable under the validity constraint, we consider the oracle regime $\hat{p} = p^*$ and assume that, given $(P_i, C_i)$, the labels $y_{i,j}$ are independent Bernoulli with mean $p_i^*(c_{i,j})$. This idealized assumption is used only to obtain a tractable benchmark: (1) it factorizes the conditional joint distribution across claims, (2) turns the coverage constraint into a product of claim-wise probabilities, and thus (3) yields an optimal valid rule that reduces to claim-wise thresholding of the oracle score. Under this simplified structure, the oracle filter has a closed-form expression maximizing retention subject to coverage and serves as the efficiency benchmark for our framework.

In this oracle setting, the resulting conformity scores become exactly uniform on $[0, 1]$ (Lemma 2 in Appendix A). Uniformity ensures that Theorem 2 achieves maximal retention efficiency: coverage

is attained precisely at the target level, without conservatism, so no true claims are unnecessarily discarded. To formalize this notion of efficiency, we introduce the *retention ratio*, which measures the proportion of claims retained under a given factuality-score function and threshold. Formally, let $(P, C, Y) \sim \mathbf{P}$ be a random document (cf. Section 3). For a factuality-score function $p$ and threshold $\tau$, define the retention ratio as

$$R(p, \tau) := \mathbb{P}\big(c \in F_\tau(p; P, C)\big). \tag{2}$$

**Theorem 3** (Retention gap with MSE). *Let $p^*$ denote the oracle factuality-score and $\hat{p}$ an estimated score. Fix a threshold $\tau \in [0, 1]$ and let $\Delta := |R(\hat{p}, \tau) - R(p^*, \tau)|$. Suppose the oracle factuality-scores satisfy a margin condition around $\tau$: there exist constants $\mathfrak{C} > 0$ and $\beta \geq 0$ such that*

$$\mathbb{P}\big(|p^*(P, c) - \tau| \leq \epsilon\big) \leq \mathfrak{C}\epsilon^\beta, \qquad \forall \epsilon > 0.$$

*Then, for any $\epsilon > 0$,*

$$\Delta \ \leq \ \frac{\mathbb{E}[(\hat{p} - p^*)^2]}{\epsilon^2} + \mathfrak{C}\epsilon^\beta.$$

*Optimizing the right-hand side over $\epsilon$ yields*

$$\Delta \leq \mathfrak{C}' \left(\mathbb{E}[(\hat{p} - p^*)^2]\right)^{\frac{\beta}{\beta+2}},$$

*where $\mathfrak{C}'$ depends only on $(\mathfrak{C}, \beta)$.*

The assumption in Theorem 3 mirrors the classical margin assumption in statistical learning (Audibert and Tsybakov, 2007, Eq. (1.7)). The exponent $\beta$ quantifies how sharply the oracle factuality-score separates cases around the threshold. When $\beta > 0$, the condition reflects a genuine margin property, whereas $\beta = 0$ corresponds to the trivial no-margin case that is included only to unify notation. Under this assumption, Theorem 3 shows that the retention gap decreases at a polynomial rate in the estimation error $\mathbb{E}[(\hat{p} - p^*)^2]$. The constant $\mathfrak{C}$ influences only the overall scale of the bound. As long as it is finite, the essential conclusion remains the same because a smaller estimation error still guarantees a smaller retention gap.

Margin-type conditions are widely used across statistical learning theory, particularly in binary classification and empirical risk minimization. Such assumptions play a central role in deriving meaningful statistical guarantees and are therefore commonly adopted in the literature.

## 4.3 Multi-LLM Ensemble

From a statistical perspective, ensembling multiple predictors reduces variance in the bias–variance tradeoff and lowers the MSE, bringing the estimator closer to the oracle benchmark. Yet directly minimizing MSE is not practical because the oracle score is unobservable and binary labels drive predictors toward overconfident outputs, which makes ensembles prone to overfitting. We therefore use a surrogate objective based on the retention decomposition. By keeping recall above a tolerance and reducing the FPR, we directly improve retention while avoiding overconfidence. This surrogate remains aligned with the oracle goal and, as our Figure 3 shows, also reduces MSE in practice.

Let $\rho := \mathbb{P}(y = 1)$ denote the marginal probability that a claim is true. The retention ratio can be decomposed as follows.

$$R(p, \tau) = \rho \cdot \text{TPR}(p, \tau) + (1 - \rho) \cdot \text{FPR}(p, \tau), \tag{3}$$

where

$$\text{TPR}(p, \tau) = \frac{\mathbb{P}(c \in F_\tau(p; P, C), y = 1)}{\rho}, \quad \text{FPR}(p, \tau) = \frac{\mathbb{P}(c \in F_\tau(p; P, C), y = 0)}{1 - \rho}.$$

Maximizing $R(p, \tau)$ therefore amounts to increasing TPR while decreasing FPR, but the two cannot be optimized simultaneously. To prevent trivial solutions that sacrifice recall, we require the true positive rate to remain above a fixed tolerance, $\text{TPR}(p, \tau) \geq 1 - \delta$ for $\delta \in (0, 1)$. With $\tau_{p,\delta}$ denoting the $\delta$-quantile of factuality-scores among true-claims, we thus focus on minimizing the FPR subject to this constraint:

$$p^\star = \arg\min_p \ \mathbb{E}\big[\text{FPR}(p, \tau_{p,\delta})\big]. \tag{4}$$

Since direct fine-tuning toward the oracle $p^*$ is impractical for black-box LLMs, we instead target the surrogate optimum $p^*$ in (4) by adopting a multi-LLM ensemble strategy. Given base factuality-scorers $\{p_m\}_{m=1}^{M}$ and weights $w = (w_1, \ldots, w_M)$, the ensemble predictor is

$$p_{\text{ens}}(P, c; w) = \sum_{m=1}^{M} w_m \, p_m(P, c),$$

with $w$ optimized to minimize the empirical FPR under the tolerance constraint (see Appendix B.1 for details). Algorithm 2 summarizes the MACI framework, which combines group-conditional conformal inference with the ensemble to maximize retention while preserving exact coverage.

## 5 EMPIRICAL RESULTS

**Dataset Structure and Evaluation Protocol.**    We empirically validate the superiority of MACI using three datasets with distinct characteristics, MedLFQA (Jeong et al., 2024; Cherian et al., 2024), WikiBio (Min et al., 2023; Cherian et al., 2024), and ExpertQA (Malaviya et al., 2024). Following prior work on false-claim filtering (Mohri and Hashimoto, 2024; Cherian et al., 2024), each example is represented as a quadruple $\{(\text{Prompt}, \text{Response}, \text{Claim Set}, \text{Ground Truth})\}$ where the response is decomposed into atomic claims and each claim is annotated with a binary factuality label. For these datasets, we define a representative grouping criterion for each benchmark and a general false-claim risk grouping. In all experiments, the underlying responses are fixed to those released with each dataset (mainly generated by GPT-4 and GPT-3.5-turbo), and every method, including CCI, BCI, and MACI, is applied to exactly the same pool of responses.

**Estimating Factuality-Score.**    To estimate factuality-score, we query an ensemble of $M$ LLMs with a verification instruction that asks them to output a verbalized factuality-score in $[0, 1]$ for each $\{(\text{Prompt}, \text{Claim})\}$ pair. In our main experiments, we use $M = 3$ models, Llama-3.3-70B-Instruct, Qwen-2.5-72B-Instruct, and DeepSeek-V3. These factuality-scores serve as base uncertainty signals. MACI aggregates the $M$ scores via the optimization-based ensemble described in Algorithm 2 to obtain a single factuality-score for each claim. Since MACI only needs access to per-claim scalar scores, it can be used as a plug-and-play filter for arbitrary generators beyond the LLMs that produced the benchmark responses.

A detailed explanation of the datasets, transformation to Log space, Fact-checking prompts (for verbalized factuality-score), group criteria, selecting LLMs, and evaluation metrics is in Appendix D. Additionally, Appendix E contains extensive additional experimental results, including comparisons with MultiValid Conformal Inference and Group Clustering methodologies, Conformity score variants, Joint probability Modeling, and discussions on MACI's operation under Covariate Shift.

### 5.1 OVERALL PERFORMANCE

Table 1 compares the group-conditional coverage and retention ratio for three datasets with distinct characteristics against two prominent baselines in the false-claim filtering field: BCI and CCI. MACI demonstrates robust performance in settings where the other two baselines falter, consistently achieving the target coverage across most groups while maintaining the highest retention ratio.

**Comparison with BCI.**    BCI only guarantees marginal coverage via a single threshold, which leads to alternating overcoverage and undercoverage depending on the difficulty differences between groups. This phenomenon is particularly evident in the results for the False-Claim Risk groups across all three datasets and the View Count groups in WikiBio. Moreover, BCI uses the conformity score of a document $D$ using a single extremal claim. This design discards potentially useful factuality information from the remaining claims and makes the document-level conformity score overly sensitive to estimation error in that single claim. Since conformal calibration will meet the target coverage, such sensitivity tends to induce conservative thresholds, which in turn filter out many true-claims. In contrast, MACI uses a multiplicative, cumulative conformity score that more directly reflects the plausibility that the entire retained set is jointly factual. By aggregating information across multiple retained claims rather than relying on a single extremal claim, MACI provides a finer-grained conformity, enabling higher retention while maintaining stable coverage.

**Comparison with CCI.**    CCI alleviates the overly conservative thresholding of BCI by employing an adaptive threshold function $g_{\text{CCI}}$ (Theorem 3.1 of Cherian et al. (2024)). It adjusts the threshold

Table 1: Group-conditional and marginal coverage, retention ratio for three datasets with distinct characteristics. The marginal results are in the row corresponding to the dataset name, followed by two rows showing the results for two representative grouping criteria for that dataset. Coverage within $1-\alpha\pm0.01$ are marked with a green dot •, while values that fall outside this range (indicating either over or undercoverage), are marked with a red arrow ↓↑. Compared to the two conformal inference baselines, MACI consistently achieves the target coverage in most cases, regardless of the group. Furthermore, its retention ratio is the highest across almost all groups. **Cov.** denotes coverage, **Ret.** denotes the retention ratio. The result with the highest retention ratio, achieved without under-coverage, is marked in **bold**. All reported values are the mean over 30 repeated trials. The performance of CCI is a result of fixing the target coverage $(1-\alpha)$.

| | Target Coverage: 80% ($\alpha = 0.2$) | | | | | | Target Coverage: 90% ($\alpha = 0.1$) | | | | | | Target Coverage: 95% ($\alpha = 0.05$) | | | | | |
| | BCI | | CCI | | MACI | | BCI | | CCI | | MACI | | BCI | | CCI | | MACI | |
| Group | Cov. | Ret. | Cov. | Ret. | Cov. | Ret. | Cov. | Ret. | Cov. | Ret. | Cov. | Ret. | Cov. | Ret. | Cov. | Ret. | Cov. | Ret. |
|---|---|---|---|---|---|---|---|---|---|---|---|---|---|---|---|---|---|---|
| **MedLFQA** | 0.80• | 0.06 | 0.81• | 0.56 | 0.80• | **0.71** | 0.90• | 0.02 | 0.90• | 0.31 | 0.90• | **0.50** | 0.95• | 0.01 | 0.95• | 0.18 | 0.95• | **0.30** |
| **Medical Content** | | | | | | | | | | | | | | | | | | |
| Info | 0.81• | 0.06 | 0.76↓ | 0.54 | 0.80• | **0.70** | 0.91• | 0.02 | 0.86↓ | 0.30 | 0.90• | **0.48** | 0.96• | 0.01 | 0.93↓ | 0.18 | 0.95• | **0.30** |
| Interpret | 0.80• | 0.07 | 0.84↑ | 0.58 | 0.79• | **0.69** | 0.89• | 0.03 | 0.93↑ | 0.33 | 0.90• | **0.47** | 0.94• | 0.01 | 0.96• | 0.21 | 0.96• | **0.26** |
| Action | 0.79• | 0.06 | 0.85↑ | 0.49 | 0.80• | **0.73** | 0.90• | 0.02 | 0.92↑ | 0.27 | 0.90• | **0.53** | 0.96• | 0.01 | 0.96• | 0.16 | 0.95• | **0.33** |
| **False-Claim Risk** | | | | | | | | | | | | | | | | | | |
| Low | 0.84↑ | 0.07 | 0.83↑ | 0.68 | 0.79• | **0.78** | 0.94↑ | 0.03 | 0.91• | 0.41 | 0.89• | **0.52** | 0.97↑ | 0.01 | 0.95• | 0.28 | 0.95• | **0.37** |
| Medium | 0.83↑ | 0.06 | 0.81• | 0.66 | 0.79• | **0.70** | 0.89• | 0.03 | 0.90• | 0.39 | 0.91• | **0.46** | 0.94• | 0.01 | 0.95• | 0.25 | 0.95• | **0.31** |
| High | 0.73↓ | 0.06 | 0.78↓ | 0.43 | 0.80• | **0.64** | 0.88↓ | 0.01 | 0.89• | 0.22 | 0.89• | **0.41** | 0.94• | 0.01 | 0.94• | 0.12 | 0.95• | **0.26** |
| **WikiBio** | 0.81• | 0.02 | 0.79• | 0.19 | 0.81• | **0.43** | 0.90• | 0.01 | 0.89• | 0.11 | 0.90• | **0.25** | 0.95• | 0.01 | 0.93↓ | 0.06 | 0.95• | **0.13** |
| **View Count** | | | | | | | | | | | | | | | | | | |
| Low | 0.74↓ | 0.03 | 0.79• | 0.18 | 0.81• | **0.36** | 0.87↓ | 0.01 | 0.88↓ | 0.11 | 0.91• | **0.21** | 0.94• | 0.01 | 0.92↓ | 0.06 | 0.96• | **0.11** |
| Medium | 0.84↑ | 0.02 | 0.78↓ | 0.19 | 0.81• | **0.46** | 0.91• | 0.01 | 0.88↓ | 0.11 | 0.91• | **0.24** | 0.95• | 0.01 | 0.92↓ | 0.06 | 0.95• | **0.12** |
| High | 0.85↑ | 0.02 | 0.81• | 0.20 | 0.81• | **0.51** | 0.91• | 0.01 | 0.92↑ | 0.12 | 0.91• | **0.24** | 0.95• | 0.01 | 0.95• | 0.07 | 0.96• | **0.12** |
| **False-Claim Risk** | | | | | | | | | | | | | | | | | | |
| Low | 0.81• | 0.03 | 0.80• | 0.21 | 0.82↑ | **0.40** | 0.90• | 0.01 | 0.90• | 0.11 | 0.90• | **0.23** | 0.95• | 0.01 | 0.93↓ | 0.07 | 0.94• | **0.17** |
| Medium | 0.81• | 0.02 | 0.78↓ | 0.19 | 0.81• | **0.42** | 0.91• | 0.01 | 0.89• | 0.11 | 0.90• | **0.25** | 0.95• | 0.01 | 0.93↓ | 0.06 | 0.95• | **0.12** |
| High | 0.81• | 0.02 | 0.79• | 0.18 | 0.81• | **0.45** | 0.89• | 0.01 | 0.88↓ | 0.11 | 0.90• | **0.28** | 0.94• | 0.01 | 0.92↓ | 0.06 | 0.96• | **0.09** |
| **ExpertQA** | 0.91↑ | 0.13 | 0.85↑ | 0.18 | 0.80• | **0.45** | 0.91• | 0.13 | 0.85↓ | 0.17 | 0.90• | **0.15** | 0.91↓ | 0.13 | 0.85↓ | 0.17 | 0.95• | **0.10** |
| **Question Domain** | | | | | | | | | | | | | | | | | | |
| Bio/Med | 0.92↑ | 0.14 | 0.86↑ | 0.18 | 0.82↑ | **0.47** | 0.92↑ | 0.14 | 0.86↓ | 0.18 | 0.92↑ | **0.22** | 0.92↓ | 0.13 | 0.86↓ | 0.18 | 0.97↑ | **0.10** |
| Tech/Sci | 0.91↑ | 0.14 | 0.86↑ | 0.17 | 0.81• | **0.44** | 0.90• | 0.13 | 0.85↓ | 0.16 | 0.89• | **0.21** | 0.91↓ | 0.13 | 0.85↓ | 0.16 | 0.94• | **0.10** |
| Common | 0.90↑ | 0.13 | 0.84↑ | 0.18 | 0.78↓ | 0.43 | 0.89• | 0.13 | 0.85↓ | 0.17 | 0.89• | **0.21** | 0.89• | 0.14 | 0.84↓ | 0.17 | 0.95• | **0.09** |
| **False-Claim Risk** | | | | | | | | | | | | | | | | | | |
| Low | 0.95↑ | 0.13 | 0.85↑ | 0.31 | 0.81• | **0.57** | 0.94↑ | 0.13 | 0.84↓ | 0.31 | 0.89• | **0.35** | 0.95• | 0.13 | 0.84↓ | 0.31 | 0.96• | **0.16** |
| Medium | 0.91↑ | 0.13 | 0.87↑ | 0.18 | 0.81• | **0.42** | 0.91• | 0.13 | 0.86↓ | 0.18 | 0.89• | **0.23** | 0.91↓ | 0.13 | 0.86↓ | 0.18 | 0.96• | **0.11** |
| High | 0.87↑ | 0.13 | 0.85↑ | 0.12 | 0.79• | **0.37** | 0.87↓ | 0.13 | 0.85↓ | 0.12 | 0.90• | **0.15** | 0.87↓ | 0.13 | 0.85↓ | 0.12 | 0.95• | **0.07** |

at the sample level, yielding a higher retention ratio. However, its conformity score for $D$ still relies on a single extremal claim. As a result, even with a well-optimized $g_{CCI}$, this single-claim–based conformity score remains sensitive to estimation error in that single extremal claim and may lead to conservative thresholds, thereby limiting retention gains. Moreover, $g_{CCI}$ operates within a linear feature-space framework, which imposes additional limitations when applying it to some grouping scenarios. The grouping criteria for each dataset, such as Medical Content or False-Claim Risk, are complex semantic functions implemented based on prompt and claim parsing. It is therefore difficult to capture such criteria using the simple linear functions and features proposed by CCI, leading to undercoverage or overcoverage. In contrast to $g_{CCI}$, our grouping function $g$ is an arbitrary measurable function that partitions the space into a finite number of groups, therefore unaffected by the complexity of grouping criteria in threshold calculations. The constraints inherent in $g_{CCI}$ are also reflected in its retention ratio. $g_{CCI}$ risks calculating an overly conservative threshold depending on how well it captures the grouping criteria. This leads to a lower retention ratio compared to MACI, which calculates group-conditional thresholds directly. CCI proposes improving retention by applying per-sample adaptive error rates that reflect each sample's characteristics. The method learns $\alpha$ as a function and lowers $\alpha$ for each sample instead of merely exceeding a minimum retention target. However, this adaptive $\alpha$ differs from our objective. Our goal is to design filtering rules that guarantee, with high probability, that the filtered set contains no false-claims, ensuring applicability in real high-stakes domains. Adapting $\alpha$ to raise retention produces filtering rules that are difficult

to deploy in such settings. Figure 2 compares CCI with adaptive $\alpha$ and MACI with $\alpha = 0.1$ on WikiBio. The upper plot sets CCI's target retention to MACI's average retention and outputs a per-sample adaptive $\alpha$, showing that CCI's $\alpha$ values are generally higher than MACI's fixed small $\alpha$. The lower table reports actual coverage and retention for both methods. CCI raises retention to nearly match MACI by increasing $\alpha$ overall, but the actual coverage is lower than MACI because the target $\alpha$ is larger.

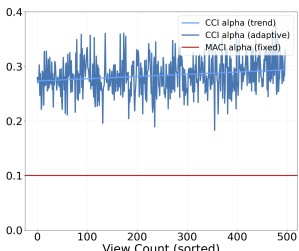

|  | CCI ($\alpha$=adap.) | | MACI ($\alpha$=0.1) | |
|---|---|---|---|---|
| **Group** | **Cov.** | **Ret.** | **Cov.** | **Ret.** |
| **WikiBio** | 0.72 | 0.26 | 0.90• | **0.28** |
| **View Count** | | | | |
| Low | 0.71 | 0.24 | 0.91• | **0.29** |
| Medium | 0.73 | 0.26 | 0.90• | **0.27** |
| High | 0.74 | 0.28 | 0.90• | **0.31** |

Figure 2: Performance comparison of CCI (adaptive $\alpha$) and MACI measured by View Count on the WikiBio dataset. The horizontal axis of the left graph is the sample index sorted by View Count, and the vertical axis is $\alpha$. The left graph shows the variation in $\alpha$ when CCI (adaptive $\alpha$) sets its target retention ratio to MACI's average retention ratio. CCI (adaptive $\alpha$) trades off higher $\alpha$ to achieve a higher retention ratio, and the table below shows the resulting decrease in coverage.

## 5.2 MULTI-LLM ENSEMBLE

In Sections 4.1, 4.2, and 4.3, we have discussed the importance of the factuality-score $\hat{p}$. Consequently, we first verify to what extent our proposed multi-LLM ensemble and optimization method (Section 4.3) improve the performance of $\hat{p}$ compared to a single-LLM, and how the retention ratio is correspondingly improved. We first find that models exhibit significant disagreements in false-claim detection. Figure 3 (a) shows the high Jaccard distance (Jaccard, 1901) between the sets of claims that different LLMs classify as false. The analysis is performed exclusively on the subset of MedLFQA claims with false ground truth. This high distance implies that the models have different patterns for detecting false-claims, suggesting significant potential for performance enhancement through an ensemble. Figure 3 (b) shows that the FPR and MSE are sequentially improved from the single-LLM to the arithmetic mean ensemble, and finally to MACI. Figure 3 (b) also shows that an improvement in FPR is consistently accompanied by an improvement in MSE, and demonstrates that MACI's $\hat{p}$ is a superior estimator of factuality-score. Figure 3 (c) demonstrates that the corresponding sequential increase in the retention ratio aligns with our objective of maximizing it by enhancing the quality of $\hat{p}$.

## 5.3 MACI UNDER COVARIATE SHIFT

In deployed systems, the calibration queries used to fit MACI's thresholds and the test-time queries need not follow the same covariate distribution. To study this covariate-shift setting, we construct an explicit shift on MedLFQA by ranking responses with a SelfCheck-based factuality score and assigning more factual easy queries to the calibration pool and more hallucination-prone hard queries to the test pool. This induces a clear mismatch between the calibration and test distributions while keeping the underlying annotation procedure fixed. Following the density-ratio correction idea of Tibshirani et al. (2019), we consider a variant, MACI-DRE, that estimates the density ratio $r(x) = p_X^{(t)}(x)/p_X^{(s)}(x)$ between test and calibration covariates using a lightweight classifier on deployment-time features (summary statistics of SelfCheck scores and prompt/response lengths). We then resample the calibration set according to the estimated ratios and run the original MACI pipeline on this resampled set, without changing any internal components of MACI. Table 2 summarizes the results on MedLFQA under the covariate shift. MACI exhibits under- or over-coverage in some false-claim risk groups, whereas MACI-DRE moves group-wise coverage closer to the target level while maintaining comparable retention. This shows that MACI can be combined with lightweight density-ratio estimation to mitigate covariate shift in practice. Detailed explanation of MACI-DRE is described in Appendix E.

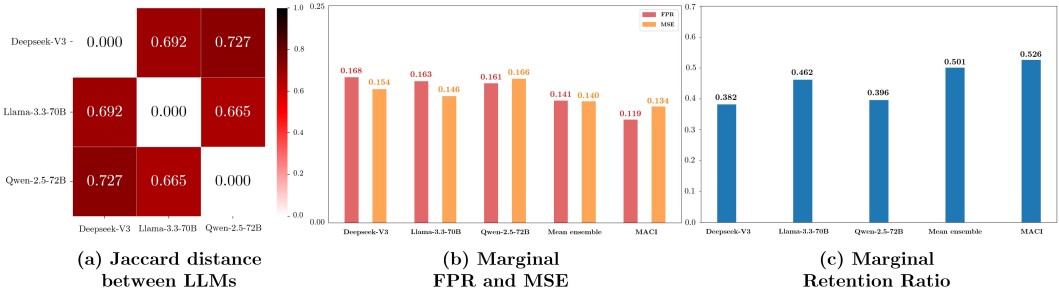

(a) Jaccard distance between LLMs

(b) Marginal FPR and MSE

(c) Marginal Retention Ratio

Figure 3: (a) shows the high Jaccard distance between different LLMs' predictions on claims known to be false in MedLFQA, indicating diverse false-claim detection patterns that support using an ensemble. (b) demonstrates the sequential improvement in FPR from a single-LLM and a simple arithmetic mean ensemble to our proposed MACI. It also demonstrates that as the FPR improves, the MSE also improves in practice; (c) demonstrates that as the FPR and MSE improve, the retention ratio also increases. A more detailed analysis is in Figure 7.

Table 2: Comparison between MACI and MACI-DRE. MACI-DRE reduces miscoverages in the scenarios where under- or over-coverage occurs by utilizing resampled calibration thresholds.

| | Target Coverage: 80% ($\alpha = 0.2$) | | | | Target Coverage: 90% ($\alpha = 0.1$) | | | | Target Coverage: 95% ($\alpha = 0.05$) | | | |
| | MACI | | MACI-DRE | | MACI | | MACI-DRE | | MACI | | MACI-DRE | |
| Group | Cov. | Ret. | Cov. | Ret. | Cov. | Ret. | Cov. | Ret. | Cov. | Ret. | Cov. | Ret. |
|---|---|---|---|---|---|---|---|---|---|---|---|---|
| **MedLFQA: False-Claim Risk** | | | | | | | | | | | | |
| Low | 0.68 | 0.83 | 0.76 | 0.72 | 0.85 | 0.57 | 0.93 | 0.35 | 0.94 | 0.34 | 0.95 | 0.29 |
| Medium | 0.65 | 0.77 | 0.84 | 0.56 | 0.82 | 0.55 | 0.94 | 0.27 | 0.87 | 0.42 | 0.96 | 0.19 |
| High | 0.77 | 0.62 | 0.75 | 0.67 | 0.88 | 0.39 | 0.89 | 0.38 | 0.92 | 0.27 | 0.93 | 0.28 |

## 5.4 TIME COST

Time efficiency is critical for real-time filtering. We compare MACI with baselines across two phases: Factuality-Score Generation, where factuality-scores are created, and Calibration, where parameters and thresholds are optimized and calculated. We exclude the negligible filtering time. Table 3 reports the costs on the WikiBio dataset. The Factuality-Score Generation phase employs Llama-3.3-70B via the OpenRouter. The results indicate that sampling-based methods suffer from high latency due to repeated response generation (SelfCheck) or knowledge graph construction (FSC-KG). In contrast, MACI achieves the lowest total wall-clock time by utilizing a single-pass scoring and a streamlined calibration process that avoids the complex parameter search of CCI.

Table 3: Time-Cost Comparison. Wall-Clock time is calculated as: calibration time + (score generation time × # test samples). Values represent the end-to-end time for 500 WikiBio test samples. Note that CCI's score generation time is identical to SelfCheck as it relies on the same sampling process, and sampling-based methods do not require a separate calibration phase.

| Phase | SelfCheck | FSC-KG | CCI | MACI |
|---|---|---|---|---|
| Factuality-Score (s) | $3.25 \pm 0.43$ | $19.30 \pm 2.81$ | $3.25 \pm 0.43$ | **$1.20 \pm 0.13$** |
| Calibration (s) | — | — | $10.33 \pm 1.18$ | **$3.24 \pm 0.65$** |
| Wall-Clock Time (s) | — | — | 1643.91 | **598.98** |

## 6 CONCLUSIONS

We reformulate conformal inference through a multiplicative filtering structure, providing a framework for false-claim filtering with finite-sample, distribution-free guarantees. Our analysis reveals how deviations from the oracle factuality-score impact retention, motivating the use of ensemble methods to narrow this gap. Building on these insights, we develop MACI, which uses ensemble-based factuality-scores and group-conditional calibration to provide group-conditional coverage guarantees. Experiments demonstrate that MACI achieves user-specified coverage while substantially improving factual claim retention and running more efficiently than existing methods, offering a practical solution for deploying LLMs in high-stakes applications.

ACKNOWLEDGMENTS

The authors gratefully acknowledges support from the National Research Foundation of Korea(NRF) grant funded by the Korea government(MSIT)(RS-2024-00457216, RS-2023-00211073).

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

APPENDIX

**Overview of Appendices.** Appendix A contains the proofs of the main theoretical results that were omitted from the paper. Appendix B provides additional methodological details, including precise definitions of the ensemble objective and empirical quantities used in our framework. Appendix C reviews background material on conformal inference and its adaptation to false-claim filtering. Appendix D reports implementation details, datasets, and evaluation metrics for our numerical experiments, together with supplementary results. Appendix E contains the additional experimental results. Appendix F reports the use of a Large Language Model for our research.

## A PROOFS OF MAIN RESULTS

**Lemma 1.** *For each $i \in \{1, \ldots, n\}$, each threshold $\tau \in [0, 1]$, and each auxiliary randomization $U_i \sim \mathrm{Unif}(0, 1)$, we have*

$$\{E_i \leq \tau\} \iff \{F(\hat{p}, \tau, U_i; P_i, C_i) \subseteq A_i\}.$$

*Proof.* Fix $i \in \{1, \ldots, n\}$, a threshold $\tau \in [0, 1]$, and a randomization variable $U_i \sim \mathrm{Unif}(0, 1)$. By the definition of $E_i$,

$$E_i = \inf \{\tau \in [0, 1] : F(\hat{p}, \tau, U_i; P_i, C_i) \subseteq A_i\}.$$

($\Rightarrow$) Suppose $E_i \leq \tau$. Then, by the definition of the infimum, there exists $\tau^* \leq \tau$ such that

$$F(\hat{p}, \tau^*, U_i; P_i, C_i) \subseteq A_i.$$

Since the retained set $F(\hat{p}, t, U_i; P_i, C_i)$ is monotone non-increasing in $\tau$, we have

$$F(\hat{p}, \tau, U_i; P_i, C_i) \subseteq A_i.$$

($\Leftarrow$) Conversely, suppose that

$$F(\hat{p}, \tau, U_i; P_i, C_i) \subseteq A_i.$$

Then $\tau$ belongs to the set

$$\{\tau \in [0, 1] : F(\hat{p}, \tau, U_i; P_i, C_i) \subseteq A_i\}.$$

Hence, by the definition of the infimum, we obtain $E_i \leq \tau$. Combining the two directions establishes the desired equivalence. That is,

$$\{E_i \leq \tau\} \iff \{F(\hat{p}, \tau, U_i; P_i, C_i) \subseteq A_i\}.$$

$\square$

### A.1 PROOF OF THEOREM 1

The proof follows the standard argument for marginal coverage in conformal prediction and is restated here in our setting.

**Lower bound.** By Algorithm 1 and Lemma 1, the event that all retained claims are factual is written as

$$\{F_{n,\alpha}(P_{n+1}, C_{n+1}) \subseteq A_{n+1}\} \iff \{E_{n+1} \leq \hat{Q}_{1-\alpha}\}.$$

Since the samples $(P_i, C_i, Y_i)$ are exchangeable and the randomizations $U_i$ are i.i.d., the conformity scores $\{E_1, \ldots, E_{n+1}\}$ are themselves exchangeable.

$$\mathbb{P}(E_{n+1} \leq \hat{Q}_{1-\alpha}) \geq 1 - \alpha.$$

Using the equivalence above, we obtain

$$\mathbb{P}(F_{n,\alpha}(P_{n+1}, C_{n+1}) \subseteq A_{n+1}) \geq 1 - \alpha,$$

which proves the marginal coverage lower bound.

**Upper bound.** Assume the conformity scores $\{E_i\}_{i=1}^{n+1}$ are distinct with probability one, eliminating the possibility of ties. Denote their order statistics by $E_{(1)}, \ldots, E_{(n+1)}$, which under this condition form a strictly increasing sequence almost surely. Let $k = \lceil (1-\alpha)(n+1) \rceil$. By construction,

$$\{F_{n,\alpha}(P_{n+1}, C_{n+1}) \subseteq A_{n+1}\} \iff \{E_{n+1} \leq E_{(k)}\}.$$

Since the conformity scores are exchangeable and distinct, the rank of $E_{n+1}$ is uniformly distributed on $\{1, \ldots, n+1\}$. It follows that

$$\mathbb{P}(E_{n+1} \leq E_{(k)}) = \frac{k}{n+1} = \frac{\lceil (1-\alpha)(n+1) \rceil}{n+1}.$$

Finally, since

$$\frac{\lceil (1-\alpha)(n+1) \rceil}{n+1} \leq 1 - \alpha + \frac{1}{n+1},$$

we conclude that

$$\mathbb{P}\big(F_{n,\alpha}(P_{n+1}, C_{n+1}) \subseteq A_{n+1}\big) \leq 1 - \alpha + \frac{1}{n+1},$$

which establishes the upper bound.

## A.2 PROOF OF THEOREM 2

Fix $k \in \{1, \ldots, K\}$ with $\mathbb{P}(g(P_{n+1}, C_{n+1}) = k) > 0$. By Algorithm 1 and Lemma 1,

$$\{F_{n,\alpha}^{(k)}(P_{n+1}, C_{n+1}) \subseteq A_{n+1}\} \iff \{E_{n+1}^{(k)} \leq \hat{Q}_{1-\alpha}^{(k)}(\{E_i^{(k)}\}_{i \in \mathcal{I}_k})\},$$

where $\mathcal{I}_k = \{i \in [n] : g(P_i, C_i) = k\}$.

Condition on $g(P_{n+1}, C_{n+1}) = k$. Then the conformity scores $\{E_i^{(k)} : i \in \mathcal{I}_k\} \cup \{E_{n+1}^{(k)}\}$ are exchangeable. Let $m = |\mathcal{I}_k|$ and set $r = \lceil (1-\alpha)(m+1) \rceil$. If $E_{(1)}^{(k)} \leq \cdots \leq E_{(m+1)}^{(k)}$ are the order statistics, then

$$\{E_{n+1}^{(k)} \leq \hat{Q}_{1-\alpha}^{(k)}(\{E_i^{(k)}\})\} \iff \{E_{n+1}^{(k)} \leq E_{(r)}^{(k)}\}.$$

By exchangeability, $E_{n+1}^{(k)}$ is equally likely to occupy any of the $m+1$ ranks. If ties occur at the cutoff, the event $\{E_{n+1}^{(k)} \leq E_{(r)}^{(k)}\}$ only becomes more likely. Therefore,

$$\mathbb{P}(E_{n+1}^{(k)} \leq E_{(r)}^{(k)} \mid g(P_{n+1}, C_{n+1}) = k) \geq \frac{r}{m+1} \geq 1 - \alpha.$$

This completes the proof.

**Lemma 2** (Uniformity under oracle factuality-score). *If $\hat{p} = p^*$, then conditionally on $(P_i, C_i)$ the conformity score $E_i$ is uniformly distributed on $[0,1]$.*

*Proof.* Recall from Section 4.1 that $F_\tau^{\text{oracle}}(P_i, C_i)$ denotes the set of retained claims at threshold $\tau$. The conformity score is defined as the smallest threshold at which the retained set is entirely factual:

$$E_i^* := \inf\{\tau \in [0,1] : F_\tau^{\text{oracle}}(P_i, C_i) \subseteq A_i\}.$$

By construction of the randomized oracle filter, the retention rule is calibrated to satisfy

$$\mathbb{P}\big(F_\tau^{\text{oracle}}(P_i, C_i) \subseteq A_i \mid P_i, C_i\big) = \tau.$$

This equality holds for every $\tau \in [0,1]$. Consequently,

$$\mathbb{P}(E_i^* \leq \tau \mid P_i, C_i) = \tau.$$

Equivalently, the conditional distribution function of $E_i$ is

$$G_{E_i^* | (P_i, C_i)}(\tau) = \tau.$$

Thus, conditional on $(P_i, C_i)$, we have $E_i^* \sim \text{Unif}(0,1)$. $\square$

### A.3 PROOF OF THEOREM 3

For notational simplicity, we suppress the dependence on $(P, c)$ and write $\hat{p} := \hat{p}(P, c)$ and $p^* := p^*(P, c)$ throughout the proof. We also emphasize that the argument below focuses on a simplified thresholding rule with a fixed threshold $\tau$. Rather than analyzing the full data-dependent and multiplicative filtering procedure used in MACI, this proof considers a basic decision rule $h_p = \mathbb{1}\{p \geq \tau\}$ to clarify how estimation error in $p$ affects the retention ratio.

Recall from (2) that the retention ratio can be written as

$$R(p, \tau) = \mathbb{P}\big(c \in F_\tau(p; P, C)\big).$$

In the thresholding case where $F_\tau(p; P, C) = \{c : p \geq \tau\}$, this simplifies to $R(p, \tau) = \mathbb{E}[h_p]$ with $h_p := \mathbb{1}\{p \geq \tau\}$. Therefore, the retention gap is

$$\begin{aligned}
\Delta &= |R(\hat{p}, \tau) - R(p^*, \tau)| \\
&= \big|\mathbb{E}[h_{\hat{p}}] - \mathbb{E}[h_{p^*}]\big| \\
&= \big|\mathbb{E}[h_{\hat{p}} - h_{p^*}]\big| \leq \mathbb{E}\big[|h_{\hat{p}} - h_{p^*}|\big],
\end{aligned}$$

where we used the inequality $|\mathbb{E}[Z]| \leq \mathbb{E}[|Z|]$.

Since $h_{\hat{p}}, h_{p^*} \in \{0, 1\}$, their absolute difference equals 1 precisely when the two thresholding decisions disagree. Hence

$$\begin{aligned}
\Delta &\leq \mathbb{P}\big(h_{\hat{p}} \neq h_{p^*}\big) \\
&= \mathbb{P}\big((\hat{p} - \tau)(p^* - \tau) < 0\big).
\end{aligned}$$

Fix $\epsilon > 0$. If $h_{\hat{p}} \neq h_{p^*}$, then one score is above $\tau$ and the other below. This can only happen in two cases:

1. $p^*$ lies within $\epsilon$ of $\tau$, i.e. $|p^* - \tau| \leq \epsilon$.
2. $p^*$ is farther than $\epsilon$ from $\tau$ but $\hat{p}$ crosses the threshold, which forces $|\hat{p} - p^*| > \epsilon$.

Therefore,

$$\{h_{\hat{p}} \neq h_{p^*}\} \subseteq \{|p^* - \tau| \leq \epsilon\} \cup \{|\hat{p} - p^*| > \epsilon\}.$$

Taking probabilities and applying the union bound gives

$$\Delta \leq \underbrace{\mathbb{P}\big(|\hat{p} - p^*| > \epsilon\big)}_{(\mathrm{I})} + \underbrace{\mathbb{P}\big(|p^* - \tau| \leq \epsilon\big)}_{(\mathrm{II})}.$$

For $(\mathrm{I})$, by Markov's inequality,

$$\mathbb{P}\big(|\hat{p} - p^*| > \epsilon\big) \leq \frac{\mathbb{E}[(\hat{p} - p^*)^2]}{\epsilon^2}.$$

For $(\mathrm{II})$, assumption (margin condition) ensures

$$\mathbb{P}\big(|p^* - \tau| \leq \epsilon\big) \leq \mathfrak{C}\epsilon^\beta.$$

Hence for every $\epsilon > 0$,

$$\Delta \leq \frac{\mathbb{E}[(\hat{p} - p^*)^2]}{\epsilon^2} + \mathfrak{C}\epsilon^\beta.$$

Let $V := \mathbb{E}[(\hat{p} - p^*)^2]$. The inequality

$$\Delta \leq \frac{V}{\epsilon^2} + \mathfrak{C}\epsilon^\beta$$

holds for any $\epsilon > 0$. Hence, we may minimize the right-hand side over $\epsilon$. Balancing the two contributions by setting $\epsilon = V^{1/(\beta+2)}$ (up to constant factors) yields

$$\Delta \leq \mathfrak{C}' V^{\frac{\beta}{\beta+2}},$$

where $\mathfrak{C}'$ depends only on $(\mathfrak{C}, \beta)$. This completes the proof.

---

**Algorithm 1** Adaptive Conformal Inference (ACI)

---

1: **Input:** Calibration dataset $\mathcal{D}_{\mathrm{cal}} = \{(P_i, C_i, Y_i)\}_{i=1}^{n_{\mathrm{cal}}}$ of size $n_{\mathrm{cal}}$, a new instance $(P_{n+1}, C_{n+1})$, a black-box classifier $\hat{p}$, and an error level $\alpha \in (0, 1)$.

2: **Output:** Filtered set $F_{n,\alpha}(P_{n+1}, C_{n+1})$ that satisfies marginal coverage.

— **Calibration Phase** —

3: **for** $i = 1, \ldots, n_{\mathrm{cal}}$ **do**
4:     Sample $U_i \sim \mathrm{Unif}(0, 1)$.
5:     Let $A_i = \{ c_{i,j} \in C_i : y_{i,j} = 1 \}$ be the set of factual claims.
6:     Compute conformity score

$$E_i = \inf\{\tau \in [0, 1] : F(\hat{p}, \tau, U_i; P_i, C_i) \subseteq A_i\}.$$

7: **end for**
8: Compute empirical quantile

$$\hat{Q}_{1-\alpha} = \inf\left\{q \in [0, 1] : \frac{1}{n_{\mathrm{cal}}} \sum_{i=1}^{n_{\mathrm{cal}}} \mathbb{1}\{E_i \leq q\} \geq 1 - \alpha\right\}.$$

— **Filtering Phase** —

9: Sample $U_{n+1} \sim \mathrm{Unif}(0, 1)$.
10: Construct the conformal filter

$$F_{n,\alpha}(P_{n+1}, C_{n+1}) = F(\hat{p}, \hat{Q}_{1-\alpha}, U_{n+1}; P_{n+1}, C_{n+1}).$$

11: **Return** $F_{n,\alpha}(P_{n+1}, C_{n+1})$.

---

# B  METHODOLOGICAL DETAILS

## B.1  DETAILS OF MULTI-LLM ENSEMBLE OBJECTIVE

This appendix provides the formal definitions of the proxy objective, empirical quantities, and optimization procedure underlying our multi-LLM ensemble (MACI), complementing the description in Section 4.3.

Recall from (3) that

$$R(p, \tau) = \rho \cdot \mathrm{TPR}(p, \tau) + (1 - \rho) \cdot \mathrm{FPR}(p, \tau).$$

Because TPR and FPR cannot be optimized simultaneously, we enforce a tolerance $\delta \in (0, 1)$ such that $\mathrm{TPR}(p, \tau) \geq 1 - \delta$. Let $\tau_{p,\delta}$ denote the $\delta$-quantile of factuality-scores among true-claims. The population-level objective is then

$$p^\star = \arg\min_p \, \mathbb{E}\big[\, \mathrm{FPR}(p, \tau_{p,\delta})\big],$$

where FPR is the false positive rate at threshold $\tau_{p,\delta}$.

Since the distribution $\mathbf{P}$ is unknown, we approximate the objective using a hold-out set $\mathcal{D}_{\mathrm{opt}} = \{(P_\ell, C_\ell, Y_\ell)\}_{\ell=1}^{n_{\mathrm{opt}}}$. Let $N_1 = \sum_{\ell=1}^{n_{\mathrm{opt}}} |\{c_{\ell,j} \in C_\ell : y_{\ell,j} = 1\}|$ be the total number of true-claims. The empirical $\delta$-quantile among true-claims is

$$\hat{\tau}_{p,\delta} = \inf\left\{t : \frac{1}{N_1} \sum_{\ell=1}^{n_{\mathrm{opt}}} \sum_{c \in C_\ell : y = 1} \mathbb{1}\{p(P_\ell, c) \leq t\} \geq \delta\right\}.$$

For document $(P_\ell, C_\ell, Y_\ell)$, the empirical FPR is defined as

$$\widehat{\mathrm{FPR}}_\ell(p, \tau) = \frac{|\{c \in F_\tau(p; P_\ell, C_\ell) : y = 0\}|}{1 \vee |\{c \in C_\ell : y = 0\}|},$$

where $a \vee b = \max(a, b)$. The empirical optimization problem is then

$$\hat{p} = \arg\min_p \, \frac{1}{n_{\mathrm{opt}}} \sum_{\ell=1}^{n_{\mathrm{opt}}} \widehat{\mathrm{FPR}}_\ell(p, \hat{\tau}_{p,\delta}).$$

---

**Algorithm 2** Multi-LLM Adaptive Conformal Inference (MACI)

---

1: **Input:** Data $\mathcal{D}_{\text{opt}} = \{(P_i, C_i, Y_i)\}_{i=1}^{n_{\text{opt}}}$ and $\mathcal{D}_{\text{cal}} = \{(P_i, C_i, Y_i)\}_{i=1}^{n_{\text{cal}}}$, of sizes $n_{\text{opt}}$ and $n_{\text{cal}}$ respectively, a new instance $(P_{n+1}, C_{n+1})$, a collection of base classifiers $\{\hat{p}_m\}_{m=1}^M$, a grouping function $g$, an error level $\alpha \in (0,1)$, and a TPR tolerance $\delta \in (0,1)$.

2: **Output:** A filtered subset $\hat{F}_{n,\alpha}^{(k_{\text{test}})}(P_{n+1}, C_{n+1})$ that satisfies group-conditional coverage.

   **— Optimization and Calibration Phase —**

3: **for** each group $k \in \{1, \ldots, K\}$ **do**
4:     Define the optimization indices $\mathcal{I}_{\text{opt},k} = \{\, i \in \mathcal{D}_{\text{opt}} : g(P_i, C_i) = k \,\}$.
5:     For any candidate weights $w$, compute the empirical threshold

$$\hat{\tau}_{\hat{p}_{\text{ens}}(w),\delta} := \inf\left\{ t \in \mathbb{R} : \frac{1}{N_{1,k}} \sum_{i \in \mathcal{I}_{\text{opt},k}} \sum_{c \in C_i : y=1} \mathbb{1}\{\hat{p}_{\text{ens}}(P_i, c; w) \leq t\} \geq \delta \right\},$$

    where $N_{1,k} = \sum_{i \in \mathcal{I}_{\text{opt},k}} |\{c_{i,j} \in C_i : y_{i,j} = 1\}|$ is the number of true-claims in group $k$.
6:     Compute the optimal ensemble weights $w_k^*$ by solving

$$w_k^* = \arg\min_w \ \frac{1}{|\mathcal{I}_{\text{opt},k}|} \sum_{i \in \mathcal{I}_{\text{opt},k}} \widehat{\text{FPR}}_i\big(\hat{p}_{\text{ens}}(w), \hat{\tau}_{\hat{p}_{\text{ens}}(w),\delta}\big)$$

$$\text{subject to} \quad \frac{1}{|\mathcal{I}_{\text{opt},k}|} \sum_{i \in \mathcal{I}_{\text{opt},k}} \widehat{\text{TPR}}_i\big(\hat{p}_{\text{ens}}(w), \hat{\tau}_{\hat{p}_{\text{ens}}(w),\delta}\big) \geq 1 - \delta.$$

7: **end for**

8: **for** each group $k \in \{1, \ldots, K\}$ **do**
9:     Define the calibration indices $\mathcal{I}_{\text{cal},k} = \{\, i \in \mathcal{D}_{\text{cal}} : g(P_i, C_i) = k \,\}$.
10:     Define the group-conditional ensemble classifier $\hat{p}_k^*(c) = \hat{p}_{\text{ens}}(c; w_k^*)$.
11:     **for** each $i \in \mathcal{I}_{\text{cal},k}$ **do**
12:         Sample $U_i \sim \text{Unif}(0,1)$.
13:         Let $A_i = \{\, c_{i,j} \in C_i : y_{i,j} = 1 \,\}$ be the set of factual claims.
14:         Compute the conformity score

$$E_i = \inf\{\tau \in [0,1] : F(\hat{p}_k^*, \tau, U_i; P_i, C_i) \subseteq A_i\}.$$

15:     **end for**
16:     Compute the group-conditional empirical quantile

$$\hat{Q}_{1-\alpha}^{(k)} = \inf\left\{ q \in [0,1] : \frac{1}{|\mathcal{I}_{\text{cal},k}|} \sum_{i \in \mathcal{I}_{\text{cal},k}} \mathbb{1}\{E_i \leq q\} \geq 1 - \alpha \right\}.$$

17: **end for**

   **— Filtering Phase —**

18: Determine the group of the new instance: $k_{\text{test}} = g(P_{n+1}, C_{n+1})$.
19: Retrieve the corresponding optimal weights $w_{k_{\text{test}}}^*$ and threshold $\hat{Q}_{1-\alpha}^{(k_{\text{test}})}$.
20: Define the group-conditional ensemble classifier $\hat{p}_{k_{\text{test}}}^*(c) = \hat{p}_{\text{ens}}(c; w_{k_{\text{test}}}^*)$.
21: Sample $U_{n+1} \sim \text{Unif}(0,1)$.
22: Construct the adaptive conformal filter:

$$\hat{F}_{n,\alpha}^{(k_{\text{test}})}(P_{n+1}, C_{n+1}) = F(\hat{p}_{k_{\text{test}}}^*, \hat{Q}_{1-\alpha}^{(k_{\text{test}})}, U_{n+1}; P_{n+1}, C_{n+1}).$$

23: **Return** $\hat{F}_{n,\alpha}^{(k_{\text{test}})}(P_{n+1}, C_{n+1})$.

---

Direct fine-tuning toward $p^*$ is infeasible in black-box LLMs. Instead, let $\{p_m\}_{m=1}^M$ denote base factuality-scores and $w = (w_1, \ldots, w_M)$ a non-negative weight vector summing to one. The en-

semble predictor is

$$p_{\text{ens}}(P, c; w) = \sum_{m=1}^{M} w_m \, p_m(P, c),$$

and the weights are optimized by

$$w^{\star} = \operatorname*{argmin}_{w} \; \frac{1}{n_{\text{opt}}} \sum_{\ell=1}^{n_{\text{opt}}} \widehat{\text{FPR}}_{\ell}(p_{\text{ens}}(\cdot; w), \widehat{\tau}_{p_{\text{ens}}(\cdot; w), \delta}).$$

## C  BACKGROUND

### C.1  CONFORMAL INFERENCE

Conformal Inference (CI) (Papadopoulos et al., 2002; Vovk et al., 2005; Lei et al., 2018; Angelopoulos and Bates, 2022) is a statistical framework that provides distribution-free uncertainty quantification for any machine learning model. Under the sole assumption that the data is exchangeable, a condition satisfied by i.i.d. data, CI generates a prediction set $C(X_{n+1})$ for a new test point $X_{n+1}$ that contains the true label $Y_{n+1}$ with a user-specified probability of at least $1 - \alpha$. This is achieved through a calibration process using a hold-out calibration dataset, $D_{\text{calib}}$. The core mechanism involves defining a non-conformity score function, $S(\cdot, \cdot)$, which measures how poorly a data point $(X_i, Y_i)$ conforms to a model's predictions. For instance, a common score for a probabilistic classifier with a score function $\hat{p}$ is $S(X_i, Y_i) = 1 - \hat{p}(Y_i \mid X_i)$, where a higher score indicates that the true label was assigned a lower probability. These scores are computed for each sample in $D_{\text{calib}}$, and a threshold $\hat{\tau}$ is determined by taking the value at the $\lceil (|D_{\text{calib}}| + 1)(1 - \alpha) \rceil$-th position in the sorted list of scores. For a new test point $X_{n+1}$, the prediction set is constructed by including all possible labels $y \in \mathcal{Y}$ whose non-conformity score does not exceed this threshold, i.e., $C(X_{n+1}) = \{ y \in \mathcal{Y} \mid S(X_{n+1}, y) \le \hat{\tau} \}$. This construction provides the powerful finite-sample marginal coverage guarantee, $\mathbb{P}(Y_{n+1} \in C(X_{n+1})) \ge 1 - \alpha$, offering a robust foundation for building reliable machine learning systems.

### C.2  FALSE-CLAIM FILTERING WITH CONFORMAL INFERENCE

Mohri and Hashimoto (2024) adapt the CI framework to filter false-claims from Large Language Model (LLM) outputs, proposing a foundational method we refer to as Basic Conformal Inference (BCI). The process begins with a set of $n$ prompts, $\{P_i\}_{i=1}^{n}$. For each prompt $P_i$, an LLM generates a response $R_i$, which is then segmented into a collection of independent claims, $C_i = \{c_{i,1}, \ldots, c_{i,N_i}\}$. Each claim $c_{i,j}$ is associated with a ground-truth binary label $y_{i,j} \in \{0, 1\}$, where $y_{i,j} = 1$ denotes a true-claim and $y_{i,j} = 0$ denotes a false-claim. Thus, each data point is a tuple $D_i = (P_i, C_i, Y_i)$, and the dataset $\{D_i\}_{i=1}^{n}$ is assumed to be drawn i.i.d. from an unknown joint distribution $\mathbf{P}$. A score function, $p$, assigns a confidence-score $p(c_{i,j})$ to each claim. This score function can be constructed in various ways, such as by directly querying an LLM (Tian et al., 2023; Guan et al., 2024) or by capturing frequency (Wang et al., 2023; Manakul et al., 2023). Their formal goal is to output a filtered set of claims, $F_{n,\alpha}(P_i, C_i) \subseteq C_i$, that contains no false-claim with user-specified error rate, i.e.,

$$\mathbb{P}\big(F_{n,\alpha}(P_{n+1}, C_{n+1}) \nsubseteq A_{n+1}\big) \le \alpha,$$

They define the filtered set as all claims whose scores exceed the calibrated global threshold $\hat{\tau}$, that is, $F_{\hat{\tau}}(P_i, C_i) := \{ c_{i,j} \in C_i : p(P_i, c_{i,j}) \ge \hat{\tau} \}$. The threshold $\hat{\tau}$ is determined by the conformal procedure. Specifically, they define a non-conformity score for each sample $(C_i, Y_i)$ as the lowest possible confidence-score threshold $\tau$ that ensures all retained claims are true :

$$S(C_i, Y_i) := \inf\{\tau \in [0, 1] : F_\tau(P_i, C_i) \subseteq A_i\}$$

This non-conformity score is computed for all samples in the calibration set $D_{\text{calib}}$. The global threshold $\hat{\tau}$ is then set to the $(1 - \alpha)$ quantile of these non-conformity scores, as detailed in Section C.1. They show that if the data samples are exchangeable, this procedure satisfies the desired probability guarantee.

Prompt: What's the difference between arthritis and reactive arthritis?
Original Claims
1. [T] Arthritis is a general term.
2. [T] Arthritis is used to describe inflammation of the joints.
3. [T] Inflammation of the joints can be caused by a variety of factors.
4. [T] Factors that can cause inflammation of the joints include age, injury, and autoimmune conditions.
5. [T] Reactive arthritis is a type of arthritis.
6. [T] Reactive arthritis occurs as a reaction to an infection.
7. [T] Reactive arthritis occurs as a reaction to an infection in another part of the body.
8. [T] Reactive arthritis occurs as a reaction to an infection in the gastrointestinal tract.
9. [T] Reactive arthritis occurs as a reaction to an infection in the genitourinary tract.
10. [T] Reactive arthritis is a condition.
11. [T] Reactive arthritis is characterized by joint pain.
12. [T] Reactive arthritis is characterized by joint swelling.
13. [F] Reactive arthritis is characterized by joint stiffness.
14. [T] Reactive arthritis is characterized by symptoms such as fever.
15. [F] Reactive arthritis is characterized by symptoms such as fatigue.
16. [T] Reactive arthritis is characterized by symptoms such as eye inflammation.
17. [T] Reactive arthritis is a form of arthritis.
18. [T] Reactive arthritis resolves on its own.
19. [T] Reactive arthritis is typically temporary.
20. [T] Reactive arthritis resolves on its own once the underlying infection is treated.
21. [T] The underlying infection needs to be treated for reactive arthritis to resolve.

**BCI**
**(Mohri and Hashimoto et al.)**

~~Arthritis is a general term~~ [T] ~~used to describe inflammation of the joints.~~ [T] ~~Inflammation of the joints can be caused by a variety of factors,~~ [T] ~~including age, injury, and autoimmune conditions.~~ [T] ~~Reactive arthritis is a type of arthritis~~ [T] ~~and a form of arthritis that occurs as a reaction to an infection in another part of the body,~~ [T] ~~such as in the gastrointestinal tract~~ [T] or the genitourinary tract. [T] Reactive arthritis is a condition [T] ~~characterized by joint pain~~ [T] ~~and joint swelling,~~ [T] ~~typically by joint stiffness.~~ [F] ~~It is also characterized by symptoms such as fever~~ [T] ~~and eye inflammation,~~ [T] ~~fatigue.~~ [F] ~~Reactive arthritis is typically temporary;~~ [T] ~~it often resolves on its own~~ [T] ~~once the underlying infection is treated,~~ [T] ~~which is a necessary step for the condition to resolve.~~ [T]

**CCI**
**(Cherian et al.)**

Arthritis is a general term [T] used to describe inflammation of the joints. [T] Inflammation of the joints can be caused by a variety of factors, [T] including age, injury, and autoimmune conditions. [T] Reactive arthritis is a type of arthritis [T] and a form of arthritis that occurs as a reaction to an infection in another part of the body, [T] ~~such as in the gastrointestinal tract~~ [T] or the genitourinary tract. [T] Reactive arthritis is a condition [T] characterized by joint pain [T] and joint swelling, [T] typically by joint stiffness. [F] It is also characterized by symptoms such as fever [T] and eye inflammation, [T] fatigue. [F] Reactive arthritis is typically temporary; [T] it often resolves on its own [T] ~~once the underlying infection is treated,~~ [T] which is a necessary step for the condition to resolve. [T]

**FSC-KG**
**(Sawczyn et al.)**

Arthritis is a general term [T] used to describe inflammation of the joints. [T] Inflammation of the joints can be caused by a variety of factors, [T] including age, injury, and autoimmune conditions. [T] Reactive arthritis is a type of arthritis [T] and a form of arthritis that occurs as a reaction to an infection in another part of the body, [T] such as in the gastrointestinal tract [T] or the genitourinary tract. [T] Reactive arthritis is a condition [T] characterized by joint pain [T] and joint swelling, [T] typically by joint stiffness. [F] It is also characterized by symptoms such as fever [T] and eye inflammation, [T] ~~fatigue.~~ [F] Reactive arthritis is typically temporary; [T] it often resolves on its own [T] once the underlying infection is treated, [T] which is a necessary step for the condition to resolve. [T]

**MACI**
**(Ours)**

Arthritis is a general term [T] used to describe inflammation of the joints. [T] Inflammation of the joints can be caused by a variety of factors, [T] including age, injury, and autoimmune conditions. [T] Reactive arthritis is a type of arthritis [T] and a form of arthritis that occurs as a reaction to an infection in another part of the body, [T] such as in the gastrointestinal tract [T] or the genitourinary tract. [T] Reactive arthritis is a condition [T] characterized by joint pain [T] and joint swelling, [T] ~~typically by joint stiffness.~~ [F] It is also characterized by symptoms such as fever [T] and eye inflammation, [T] ~~fatigue.~~ [F] Reactive arthritis is typically temporary; [T] it ~~often resolves on its own~~ [T] once the underlying infection is treated, [T] which is a necessary step for the condition to resolve. [T]

Figure 4: An example of independently decomposed claims in MedLFQA and the aggregated results of four methods that filter the false-claims of those claims. BCI yields conservative results, while CCI and FSC-KG show high retention but fail to filter out all false-claims, whereas MACI successfully filters out all false-claims.

## D   EXPERIMENT DETAILS

**Transformation to Log Space.**   Our implementation performs all multiplicative computations in log-space. Given a claim-level factuality-score $\hat{p}(P, c) \in [0, 1]$ and a small $\epsilon > 0$, we apply the complement-based transform

$$\ell(P, c) := -\log(1 - \hat{p}(P, c) + \epsilon).$$

Let $\pi$ denote the ordering induced by $\hat{p}$ (ascending, so low-confidence claims are considered first), and define the complement-product

$$\Pi_k(P, C) := \prod_{j=1}^{k} \big(1 - \hat{p}(P, c_{\pi(j)}) + \epsilon\big).$$

Then we have the exact log-product identity

$$-\log \Pi_k(P, C) \ = \ \sum_{j=1}^{k} \ell(P, c_{\pi(j)}).$$

Hence the budget rule used in our filter, $\sum_{j=1}^{k} \ell(P, c_{\pi(j)}) \leq \tau$, is equivalent to $\Pi_k(P, C) \geq e^{-\tau}$. Because the mapping $x \mapsto -\log x$ is strictly monotone on $(0, 1]$, this implementation of log-space preserves the induced selection sets and conformal quantiles, while improving numerical stability by avoiding product underflow.

### D.1   DATASETS

**MedLFQA.**   For the medical question-answering task, Cherian et al. (2024) create an experimental dataset using prompts from the MedLFQA benchmark (Jeong et al., 2024). To generate the data, they first prompt GPT-3.5-Turbo to produce new responses, which are then parsed into atomic claims by GPT-4o. For the crucial step of ground-truth annotation, they employ an automated verification procedure. For each generated claim, they prompt GPT-3.5-Turbo to verify whether it is substantiated by the reference answer provided in the original MedLFQA benchmark, effectively treating the reference answer as the ground-truth source text. From this dataset, we randomly extracted 2,000 samples, comprising 33,833 claims, for our experiments.

**WikiBio.** Cherian et al. (2024) follow the principles of the FACTSCORE (Min et al., 2023) dataset to construct a new, large-scale benchmark for evaluating the factuality of LLM output. To generate the data, they prompt GPT-3.5-Turbo to write short biographies for 8,516 names sampled from Wikipedia. To circumvent the high cost of manual annotation, they then employ a variant of the FACTSCORE procedure for fact-checking. For each generated claim, they use the BM25 algorithm (Robertson and Zaragoza, 2009) to retrieve ground-truth passages from Wikipedia and subsequently prompt GPT-3.5-Turbo to verify whether the claim is supported by the retrieved text. This completed dataset is referred to as WikiBio in our paper for convenience. We randomly extracted 2,000 samples, comprising 53,804 claims, for our experiments.

**ExpertQA.** Malaviya et al. (2024) construct the ExpertQA dataset, a large-scale benchmark for evaluating the factuality and attribution of LLM output. To generate the data, they first asked 484 qualified experts across 32 fields to formulate challenging, information-seeking questions from their professional lives. To ensure high-quality annotations, they then employed an expert-in-the-loop evaluation procedure where the same experts validated sentence-level claims in responses generated by six representative LLMs. Using the rich, human-annotated information provided in this dataset, we construct a binary ground truth for each claim. Among the datasets in our study, ExpertQA was the most challenging and also the most rigorously labeled, due to its direct validation by domain experts. We randomly extracted 2,000 samples, comprising 11,538 claims, for our experiments.

Of the 2,000 samples, 1,500 were used for the calibration phase, and 500 were used for the filtering (test) phase. Figure 4 shows an actual format of the data sample we use.

## D.2 GROUPING CRITERIA

We employ complex grouping criteria that are likely to occur in reality, yet simultaneously require prompt and response parsing along with numeric values. We create a grouping criterion applicable to all three datasets and three classification criterion reflecting the characteristics of each dataset. The criteria are as follows:

**Common: False-Claim Risk.** This is a composite risk index calculated by analyzing features of the prompt and response texts. The risk score increases with longer response lengths, a higher frequency of lists or numbers, and the inclusion of absolute or definitive expressions like 'always', 'never', or 'cure'. Conversely, the risk score decreases when expressions citing sources or evidence, such as 'according to' or 'research shows', are present. This index estimates the potential risk of containing false information based solely on textual characteristics.

**MedLFQA: Medical Content.** Medical-related questions are classified into three groups based on the 'intent' of the user's prompt:
Information-Seeking (Info): Cases that ask for factual information about a specific disease or drug, using keywords like "what is," "symptom," or "treatment."
Interpretation-Seeking (Interpret): Cases that request an interpretation of what a specific symptom or condition means, using phrases like "what does it mean" or "should I worry."
Action-Seeking (Action): Cases that ask for specific guidance on actions or treatment, using phrases like "should I," "can I take," or "how to."

**WikiBio: View Count.** Groups are divided based on the cumulative number of page views for each person's Wikipedia page in the WikiBio dataset. This is used as an indicator of public interest in or awareness of the person.

**ExpertQA: Question Domain.** Questions (prompts) from the ExpertQA dataset are classified into three high-level domains based on the academic field specified in the official metadata:
Biology/Medicine (Bio/Med): Life science and health-related fields such as Healthcare, Medicine, Biology, Chemistry, and Psychology.
Technology/Science (Tech/Sci): Engineering and physics-related fields such as Engineering and Technology, Physics, and Astronomy.
Common: All other academic fields that do not fall into the two categories above.

## D.3 FACT-CHECKING PROMPT

The prompt below is an example of the prompt we input to $M$ LLMs to obtain a verbalized factuality-score for each claim.

---

**Fact-checking prompt**

**Instruction.** You are an expert fact-checker and logician with access to a vast knowledge base. Your task is to assess the factual accuracy of a specific claim extracted from a model's response to a given prompt. Follow these steps to produce a precise evaluation:

1. **Contextual Analysis**: Understand the claim within the context of the original prompt.
2. **Knowledge Verification**: Using your general world knowledge and the information provided, check the claim against established facts, scientific or medical consensus when applicable, and basic logical consistency.
3. **Probability Estimation**: Estimate the probability that the claim is factually true.

**Scoring rubric (probability of truthfulness).** Assign a continuous score between $[0.0, 1.0]$ representing your confidence:

- **1.0 (Certain Truth)**: The claim is axiomatically true or verified by very strong evidence.
- **0.8–0.9 (High Confidence)**: The claim is widely accepted as true by experts; minor context might be missing but the core is correct.
- **0.4–0.6 (Ambiguous/Uncertain)**: The claim is debated, only partially supported, or there is insufficient information to verify it.
- **0.1–0.3 (Low Confidence)**: The claim conflicts with available evidence or contains significant inaccuracies.
- **0.0 (False)**: The claim is demonstrably false, logically impossible, or entirely hallucinated.

**Input.**
**Original prompt:** {prompt_text}
**Claim to evaluate:** {claim_text}
**Output format.** Return a single JSON object with an array `"evaluations"`:

```
{
  "evaluations": [
    {
      "claim_id": 1,
      "reasoning": "Step-by-step reasoning here...",
      "score": 0.85
    }
  ]
}
```

---

## D.4 SELECTING LLMS

Although our methodology is model-free and works with any large language model, we choose to use Llama-3.3-70B-Instruct (Grattafiori et al., 2024), Qwen-2.5-72B-Instruct (Qwen et al., 2025), and DeepSeek-V3 (DeepSeek-AI et al., 2025). We selected these three white-box models for their high transparency and reproducibility. Their public availability and stable serving allow us to openly share and control all settings, such as decoding parameters, logs, and the calibration pipeline, making our work easily replicable. This open nature also reduces our dependence on unseen changes to policies or filters that come with version updates, which is a key advantage in fields where reproducibility and auditing are crucial.

## D.5 SAMPLING-BASED METHODS

**SelfCheck.** Manakul et al. (2023) proposes a black-box, zero-resource method that detects hallucinations by sampling multiple responses from a large language model for the same query and quantifying the content consistency between the original response and the samples. Specifically, the method generates multiple stochastic responses for a single prompt and calculates response reliabil-

ity by aggregating mutual consistency at the sentence/passage level, using metrics such as semantic similarity, NLI-based contradiction signals, and question-answering agreement. In our experiments, we use Llama-3.3-70B-Instruct as the model to generate multiple samples for the same prompt when applying the SelfCheck procedure.

**FactSelfCheck (FSC).** Sawczyn et al. (2025) proposes a method for detecting fact-level hallucinations by extracting factual units from a response and multiple samples to construct a fact graph, then aggregating supporting and contradictory signals for each fact across all samples. The procedure involves extracting facts (e.g., entity-relation-entity triplets) from an initial response and multiple samples, calculating the degree of consensus among these facts to aggregate them into fact, sentence, or passage-level scores, and finally performing threshold-based filtering. In our experiments, we also use Llama-3.3-70B-Instruct to generate the multiple samples required for the FSC procedure.

### D.6 EVALUATION METRICS

To evaluate our proposed method, we assess two key aspects: the quality of our oracle-approximating factuality-score function and the performance of the final filtering procedure.

**Coverage.** Coverage is the primary metric for verifying the theoretical guarantee of our conformal inference procedure. A sample $D_i$ is considered "covered" if its filtered set $F(C_i)$ contains no hallucinatory claims. The empirical coverage is the fraction of samples in the test set that are successfully covered. For a given error rate $\alpha$, a valid conformal procedure is expected to yield an empirical coverage rate approaching or exceeding $1 - \alpha$.

$$\textbf{Cov.} = \frac{1}{|\mathcal{D}_{\text{test}}|} \sum_{i \in \mathcal{D}_{\text{test}}} \mathbb{1}\big[j \in [N_i] : c_{i,j} \in F(C_i), y_{i,j} = 1\big].$$

**Retention Ratio.** While coverage measures the safety of the filter, retention ratios measure its utility. Retention measures the average fraction of total claims remaining after filtering, indicating how much of the original text volume is preserved:

$$\textbf{Ret.} = \frac{1}{|\mathcal{D}_{\text{test}}|} \sum_{i \in \mathcal{D}_{\text{test}}} \frac{|F(C_i)|}{|C_i|}.$$

## E ADDITIONAL RESULTS

**Comparison with BCI, CCI, MACI, with Unified Factuality-Score.** Theorem 3 demonstrates that the quality of the factuality-score directly impacts MACI's retention ratio. Therefore, improving the quality of the factuality-score through a Multi-LLM-based weighted optimization ensemble directly contributes to the high retention ratio compared to MACI's conformal inference-based baseline. Consequently, it is meaningful to verify how much the adaptive conformity score structure, which mimics oracle factuality by excluding the Multi-LLM ensemble part in MACI, contributes to the retention ratio. Therefore, we fix MACI's factuality-score to a frequency score, which BCI and CCI use. That is, we compare the coverage and retention of MACI with BCI and CCI, excluding the Multi-LLM ensemble component. Table 4 shows that MACI achieves the highest retention even under the same unified factuality-score. This demonstrates that MACI's oracle-motivated adaptive conformal inference structure itself is superior to the baselines.

**Comparison with MultiValid Conformal Inference.** Research in the Multivalid Conformal Prediction family presents a robust framework that simultaneously guarantees coverage for multiple subgroups and threshold intervals. Specifically, Jung et al. (2023)'s Batch Multivalid Conformal Prediction defines the concept of multivalid coverage, which simultaneously satisfies group-conditional coverage for multiple groups and threshold-level-conditional coverage for various threshold levels under a single threshold function. Formally, letting $s(X, Y)$ denote a conformity score, $\Gamma_\tau(X)$ the prediction set induced by a threshold function $\tau(\cdot)$, and $\mathcal{G}, \mathcal{I}$ denote a family of groups and score intervals, multivalid coverage requires that for all $G \in \mathcal{G}$ and $I \in \mathcal{I}$,

$$\mathbb{P}\big(Y \in \Gamma_\tau(X) \,\big|\, X \in G, \, s(X, Y) \in I\big) \geq 1 - \alpha - \varepsilon,$$

Table 4: Comparison with conformal baselines, with unified factuality-score. Coverage within $1 - \alpha \pm 0.01$ are marked with a green dot •, while values that fall outside this range are marked with a red arrow ↓↑.

| | Target Coverage: 80% ($\alpha = 0.2$) | | | | | | Target Coverage: 90% ($\alpha = 0.1$) | | | | | | Target Coverage: 95% ($\alpha = 0.05$) | | | | | |
|---|---|---|---|---|---|---|---|---|---|---|---|---|---|---|---|---|---|---|
| | BCI | | CCI | | MACI | | BCI | | CCI | | MACI | | BCI | | CCI | | MACI | |
| Group | Cov. | Ret. | Cov. | Ret. | Cov. | Ret. | Cov. | Ret. | Cov. | Ret. | Cov. | Ret. | Cov. | Ret. | Cov. | Ret. | Cov. | Ret. |
| **MedLFQA: False-Claim Risk** | | | | | | | | | | | | | | | | | | |
| Low | 0.84↑ | 0.07 | 0.83↑ | 0.68 | 0.82↑ | **0.70** | 0.94↑ | 0.03 | 0.91• | 0.41 | 0.91• | **0.49** | 0.97↑ | 0.01 | 0.95• | 0.28 | 0.96• | **0.32** |
| Medium | 0.83↑ | 0.06 | 0.81• | 0.66 | 0.81• | **0.71** | 0.89• | 0.03 | 0.90• | 0.39 | 0.91• | **0.46** | 0.94• | 0.01 | 0.95• | 0.25 | 0.96• | **0.30** |
| High | 0.73↓ | 0.06 | 0.78↓ | 0.43 | 0.81• | **0.58** | 0.88↓ | 0.01 | 0.89• | 0.22 | 0.90• | **0.36** | 0.94• | 0.01 | 0.94• | 0.12 | 0.95• | **0.24** |

for a small tolerance $\varepsilon \geq 0$. This is a stronger requirement than standard group-conditional coverage, which only conditions on group membership.

While this approach provides a strong form of validity, it can lead to a conservatively large prediction set due to the need to satisfy many groups and threshold levels simultaneously. MACI targets a more specific application setting. MACI's focus is on performing false-claim filtering for responses generated by LLMs, aiming to maintain group-conditional coverage for a small number of meaningful groups (e.g., groups based on false-claim risk) while achieving a level of retention suitable for practical systems. Concretely, if $\mathcal{G}_{\text{risk}}$ denotes a task-specific partition (e.g., low/medium/high false-claim risk), MACI enforces

$$\mathbb{P}\big(Y \in C(X) \,\big|\, Z \in g\big) \geq 1 - \alpha, \qquad \forall g \in \mathcal{G}_{\text{risk}},$$

without additionally conditioning on score intervals. Therefore, MACI focuses on balancing coverage and efficiency for the specific task of false-claim filtering, rather than providing general multi-valid guarantees across diverse groups and threshold intervals.

To quantitatively assess this difference, we implemented MultiValid Conformal Inference (MVCI), a modification of Jung et al. (2023)'s BatchMVP tailored for the false-claim filtering environment, and compared it with MACI. Experiments were conducted on the WikiBio dataset, with both methods configured to use the same group definitions (e.g., group partitioning based on false-claim risk), the same calibration/test split, and the same target error rate $\alpha$.

Table 5 shows the results comparing the group-conditional coverage and retention of the two methods. MVCI satisfies the target coverage level in each group but achieves retention at a significantly lower rate by producing conservative thresholds to simultaneously guarantee coverage across groups and threshold levels. In contrast, MACI achieves a much higher retention ratio than MVCI while maintaining a similar level of group-conditional coverage under the same group definitions and significance level. This reflects the different design philosophies: MVCI prioritizes strong multivalid coverage, whereas MACI emphasizes balancing group-conditional guarantees with practical retention in false-claim filtering.

Table 5: Comparison with MultiValid Conformal Inference (MVCI). Coverage within $1 - \alpha \pm 0.01$ are marked with a green dot •, while values that fall outside this range are marked with a red arrow ↓↑.

| | Target Coverage: 80% ($\alpha = 0.2$) | | | | Target Coverage: 90% ($\alpha = 0.1$) | | | | Target Coverage: 95% ($\alpha = 0.05$) | | | |
|---|---|---|---|---|---|---|---|---|---|---|---|---|
| | MVCI | | MACI | | MVCI | | MACI | | MVCI | | MACI | |
| Group | Cov. | Ret. | Cov. | Ret. | Cov. | Ret. | Cov. | Ret. | Cov. | Ret. | Cov. | Ret. |
| **WikiBio: False-Claim Risk** | | | | | | | | | | | | |
| Low | 0.80• | 0.11 | 0.82↑ | **0.40** | 0.89• | 0.03 | 0.90• | **0.23** | 0.95• | 0.02 | 0.94• | **0.17** |
| Medium | 0.81• | 0.08 | 0.81• | **0.42** | 0.91• | 0.02 | 0.90• | **0.25** | 0.96• | 0.01 | 0.95• | **0.12** |
| High | 0.81• | 0.03 | 0.81• | **0.45** | 0.90• | 0.01 | 0.90• | **0.28** | 0.95• | 0.01 | 0.96• | **0.09** |

**Comparison with sampling-based methods.** While our primary focus is on CI-based baselines, it is also practical to compare against recent non-CI approaches. We compare MACI's group-conditional coverage, marginal coverage, and retention ratio with sampling-based methods that apply to black-box LLMs and do not rely on retrieval. Brief descriptions of these baselines appear in Section D.5. Unlike CI-based methods, sampling-based approaches do not provide statistical guarantees; instead, they compute a factuality-score $p \in [0, 1]$, enabling false-claim filtering via a

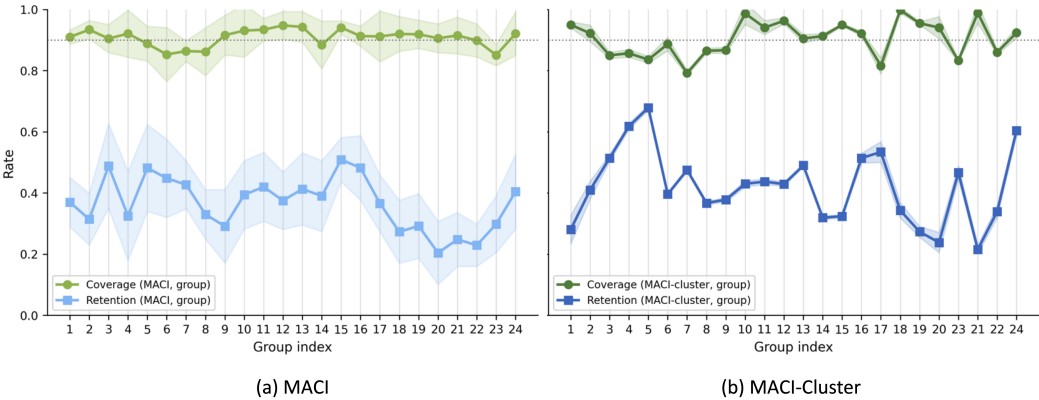

| (a) MACI | (b) MACI-Cluster |

Figure 5: Coverage and retention in large number of groups, using MACI and the MACI-cluster method. The top shows results split into 24 groups. The average coverage per group is more maintained by MACI than MACI-cluster, but with greater variance. Using the group clustering method allows for practical results by sacrificing strict group-conditional coverage.

threshold (e.g., 0.5). Table 6 shows that sampling-based methods attain high retention but low coverage. This highlights their limitation in meeting the strict requirement that the filtered set contain no false-claims. Moreover, their target coverage is not user-specified and thus unpredictable. These points underscore the need for MACI in high-stakes settings that require a user-specified high $1 - \alpha$.

Table 6: Comparison with sampling-based methods, a representative black-box and non-retrieval approach for false-claim filtering. Sampling-based methods generally exhibit very low or unstable coverage and a high retention ratio. This suggests they are unsuitable for the strict target that all claims have to be factual (the definition of **Cov.**). In contrast, MACI ($\alpha = 0.1$) demonstrates the ability to reliably guarantee the user's desired coverage.

| | SelfCheck | | FSC-text | | FSC-KG | | MACI | | | SelfCheck | | FSC-text | | FSC-KG | | MACI | |
|---|---|---|---|---|---|---|---|---|---|---|---|---|---|---|---|---|---|
| **Group** | Cov. | Ret. | Cov. | Ret. | Cov. | Ret. | Cov. | Ret. | **Group** | Cov. | Ret. | Cov. | Ret. | Cov. | Ret. | Cov. | Ret. |
| **MedLFQA** | 0.56 | 0.97 | 0.63 | 0.85 | 0.64 | 0.88 | **0.90** | 0.50 | **WikiBio** | 0.12 | 0.97 | 0.33 | 0.77 | 0.37 | 0.70 | **0.90** | 0.25 |
| **Medical Content** | | | | | | | | | **View Count** | | | | | | | | |
| Info | 0.49 | 0.98 | 0.59 | 0.85 | 0.58 | 0.87 | **0.90** | 0.48 | Low | 0.11 | 0.95 | 0.42 | 0.69 | 0.46 | 0.64 | **0.91** | 0.21 |
| Interpret | 0.54 | 0.98 | 0.59 | 0.90 | 0.63 | 0.93 | **0.90** | 0.47 | Medium | 0.13 | 0.97 | 0.31 | 0.79 | 0.27 | 0.73 | **0.91** | 0.24 |
| Action | 0.64 | 0.95 | 0.74 | 0.79 | 0.73 | 0.83 | **0.90** | 0.53 | High | 0.11 | 0.98 | 0.26 | 0.82 | 0.39 | 0.73 | **0.91** | 0.24 |
| **False-Claim Risk** | | | | | | | | | **False-Claim Risk** | | | | | | | | |
| Low | 0.64 | 0.98 | 0.70 | 0.86 | 0.71 | 0.89 | **0.89** | 0.52 | Low | 0.19 | 0.98 | 0.41 | 0.78 | 0.43 | 0.72 | **0.90** | 0.23 |
| Medium | 0.56 | 0.98 | 0.63 | 0.88 | 0.60 | 0.92 | **0.91** | 0.46 | Medium | 0.08 | 0.96 | 0.28 | 0.74 | 0.32 | 0.68 | **0.90** | 0.25 |
| High | 0.41 | 0.97 | 0.55 | 0.82 | 0.58 | 0.85 | **0.89** | 0.41 | High | 0.08 | 0.97 | 0.30 | 0.78 | 0.35 | 0.70 | **0.90** | 0.28 |

**Applying Group Clustering Method.** MACI adopts a Mondrian-style group-conditional conformal scheme: for each pre-defined group $g \in \mathcal{G}$, we calibrate a separate threshold from the calibration examples belonging to that group. Let $R_g$ denote the nonconformity score distribution within group $g$, and let $\hat{q}_g^{1-\alpha}$ be the empirical $(1 - \alpha)$-quantile of the calibration scores in $g$. Using this group-specific quantile as the threshold yields an exact finite-sample group-conditional guarantee

$$\mathbb{P}(Y \in C_g(X) \,|\, Z = g) \geq 1 - \alpha, \qquad \forall g \in \mathcal{G},$$

under group-conditional exchangeability between calibration and test examples. However, when the calibration sample size $n_g$ of a particular group is small, the empirical quantile $\hat{q}_g^{1-\alpha}$ can have high variance, which in turn leads to unstable thresholds and increased variability in both coverage and retention across groups.

Gao et al. (2025) propose to mitigate this issue by clustering groups whose score distributions are similar and then performing conformal calibration at the cluster level. Formally, let $h : \mathcal{G} \to \mathcal{K}$ map each group $g$ to a cluster $k = h(g)$, and let $R_k$ denote the score distribution obtained by pooling all calibration scores from the groups assigned to cluster $k$. Mondrian conformal inference is then applied in the cluster space, producing a threshold $r_k^\alpha$ that is shared by all groups $g$ with $h(g) = k$.

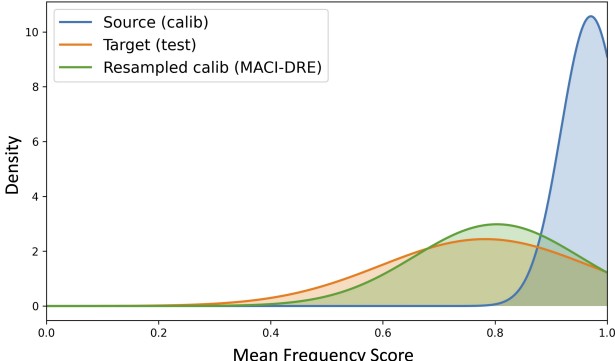

Logistic Classifier: AUC=0.943 ACC=0.894 F1=0.890

Figure 6: Covariate shift and resampling distribution in the MedLFQA dataset. The blue and red graphs show the distributions of calibration and test, respectively, based on the covariate shift variable mean frequency score. The green graph shows the distribution resampled using a binary classifier.

This increases the effective calibration sample size for each threshold and reduces variance. At the same time, the group-conditional guarantee is relaxed: if the score distribution of each group $g$ is close to that of its cluster $k = h(g)$ in the sense that

$$\sup_{t \in \mathbb{R}} \big| F_g(t) - F_k(t) \big| \ \le \ \varepsilon_g,$$

where $F_g$ and $F_k$ are the CDFs of $R_g$ and $R_k$, then the group-conditional coverage degrades from the exact $1 - \alpha$ level to

$$\mathbb{P}\big\{Y \in C_k(X) \,\big|\, Z = g\big\} \ \ge \ 1 - \alpha - \varepsilon_g,$$

for all $g \in \mathcal{G}$, as shown in Gao et al. (2025). In other words, group clustering trades strict finite-sample group-conditional coverage for lower variance and more stable thresholds.

We construct a variant, MACI-Cluster, by combining Gao et al.'s group clustering mechanism with MACI's factuality-score and thresholding scheme. We first define fine-grained groups using the false-claim risk score, apply group clustering in the space of conformity score distributions, and then learn cluster-level thresholds that are shared by the groups within each cluster. We then compare MACI-Cluster with the original MACI in terms of empirical group-conditional coverage and retention across many groups. Concretely, MACI-Cluster uses $K = 8$ clusters obtained by $k$-means on vectorized conformity-score histograms, and applies a single conformal threshold per cluster.

Figure 5 shows the coverage and retention per group for MACI and MACI-Cluster under 24 group partitions. Over 100 repeated runs, MACI's group-wise coverage is on average close to the user-specified level but exhibits large variance, making the realized coverage unstable. MACI-Cluster yields group-wise coverage that does not exactly converge to the target level due to distributional differences between clusters and evaluation groups, but its variance is smaller, leading to substantially more stable coverage and retention when the number of groups is large and each group contains few samples. Consequently, the group clustering method is a good approach that provides stable results when the number of groups increases and the sample size is small.

**MACI under Covariate Shift.** The calibration data used to fit conformal thresholds and the queries encountered at test time do not necessarily follow the same covariate distribution in deployed systems. This setting is typically formalized as covariate shift: calibration data $\{(X_i, Y_i)\}_{i=1}^{n} \overset{i.i.d.}{\sim} \mathbf{P}_{XY}^{(s)}$ and the test point $(X_{n+1}, Y_{n+1}) \sim \mathbf{P}_{XY}^{(t)}$ satisfy $\mathbf{P}_{Y|X}^{(s)} = \mathbf{P}_{Y|X}^{(t)}$, while their marginal covariate distributions $\mathbf{P}_X^{(s)}$ and $\mathbf{P}_X^{(t)}$ may differ. Let $p_X^{(s)}$ and $p_X^{(t)}$ denote the densities of $\mathbf{P}_X^{(s)}$ and $\mathbf{P}_X^{(t)}$, respectively (with respect to a common dominating measure). Tibshirani et al. (2019) address this

mismatch by reweighting calibration examples using the density ratio

$$r(x) \; = \; \frac{p_X^{(t)}(x)}{p_X^{(s)}(x)}.$$

Let $S_i = S(X_i, Y_i)$ denote nonconformity scores on the source calibration set. A weighted empirical quantile for a target miscoverage level $\alpha$ is then defined by

$$\widehat{q}_\alpha \; = \; \inf\Big\{ q \; : \; \frac{\sum_{i=1}^n r(X_i)\,\mathbb{1}\{S_i > q\}}{\sum_{i=1}^n r(X_i)} \; \le \; \alpha \Big\},$$

which replaces the usual unweighted empirical tail probability by a density-ratio weighted version that targets $\mathbf{P}_t$ instead of $\mathbf{P}_s$.

We adopt this density-ratio correction principle for the MACI pipeline and refer to the resulting variant as MACI-DRE. Rather than modifying the quantile computation inside MACI, we first use an estimate $\widehat{r}(x)$ to construct a density-ratio–weighted calibration set by importance resampling, and then run the original MACI algorithm on this resampled set. Concretely, given calibration samples $\{(X_i, Y_i)\}_{i=1}^n$ from $P_s$ and estimated ratios $\widehat{r}(X_i)$, we draw indices

$$I_1, \ldots, I_n \; \sim \; \mathrm{Multinomial}(n; \; \pi_1, \ldots, \pi_n), \qquad \pi_i \; = \; \frac{\widehat{r}(X_i)}{\sum_{j=1}^n \widehat{r}(X_j)},$$

and form a resampled calibration set $\{(X_{I_k}, Y_{I_k})\}_{k=1}^n$. For any bounded measurable function $f$,

$$\mathbb{E}\Big[\frac{1}{n}\sum_{k=1}^n f(X_{I_k}, Y_{I_k})\Big] \; = \; \sum_{i=1}^n \pi_i f(X_i, Y_i) \; \approx \; \mathbb{E}_{(X,Y)\sim P_t}\big[f(X,Y)\big] \quad \text{if} \quad \widehat{r}(x) \approx \frac{p_t(x)}{p_s(x)}.$$

Thus, when the density ratio is well estimated, running MACI on the resampled calibration set is equivalent, in expectation, to running MACI on a calibration set drawn directly from the target distribution $P_t$, while keeping all internal components of MACI (ensemble weight learning, subgroup-wise threshold estimation) unchanged.

For our covariate-shift situation on MedLFQA, we construct an explicit covariate shift between the calibration and test distributions. For each data sample $\{(P, R, C)\}$, we obtain claim-level factuality-scores via SelfCheck (Manakul et al., 2023) method, and define a scalar feature

$$s(x) \; = \; \frac{1}{m(x)} \sum_{j=1}^{m(x)} \hat{p}_{\mathrm{SelfCheck}}(x, j),$$

where $m(x)$ is the number of atomic claims in the response. We sort all samples by $s(x)$ and designate the upper region (larger $s(x)$, on average more factual responses) as the source test pool $P_s$, and the lower region (smaller $s(x)$, more hallucination-prone responses) as the target calibration pool $P_t$. This produces a clear marginal covariate shift in terms of the distribution of $s(x)$, while keeping the underlying annotation procedure fixed.

The density ratio is estimated from features that are observable at deployment time and do not require ground-truth labels. For each sample $x$, we build a feature vector

$$\phi(x) \; = \; \big(\overline{s}(x), \mathrm{std}(s(x)), \mathrm{len}(\mathrm{prompt}), \mathrm{len}(\mathrm{response}), 1\big),$$

where $\overline{s}(x)$ and $\mathrm{std}(s(x))$ denote the mean and standard deviation of the claim-level SelfCheck scores, and the remaining components encode simple prompt and response length statistics. We then train a binary classifier on $\phi(x)$ to distinguish source from target samples, with label 0 for $P_s$ and 1 for $P_t$. Using logistic regression with approximately balanced priors, the estimated density ratio is obtained as

$$\widehat{r}(x) \; = \; \frac{\hat{p}(Y = 1 \mid \phi(x))}{1 - \hat{p}(Y = 1 \mid \phi(x))},$$

which is then used for the importance resampling step described above.

**Comparison with Conformity Score Variants.** MACI's multiplicative score is motivated by the oracle formulation. Under an ideal oracle that assigns a joint factuality-score to each claim, combining the scores of multiple verifiers by multiplication provides a simple approximation to this joint score. For numerical stability, we implement this in the following log-product form:

$$S_{\text{mult}}(c) = \sum_{i=1}^{N_i} \log p_i(c).$$

Conformal calibration depends on the ranking of conformity scores rather than their absolute values, so the product and log-product forms are related by a monotone transformation and yield the same coverage guarantees.

We compare this multiplicative score against two alternative aggregation rules. The first is a log-sum form,

$$S_{\text{log-sum}}(c) = \log\Big(\frac{1}{N_i} \sum_{i=1}^{N_i} p_i(c)\Big).$$

The second is a power-mean form,

$$S_{\text{PM}}(c; \lambda) = \Big(\frac{1}{N_i} \sum_{i=1}^{N_i} p_i(c)^\lambda\Big)^{1/\lambda},$$

which allows us to explore different aggregation behaviors by varying the exponent $\lambda$.

Table 7 reports the coverage and retention of these two variants and MACI. Overall, all three methods stay close to the target coverage, but power-mean consistently achieves the lowest retention across settings. The log-sum form attains retention comparable to MACI in the low- and medium-risk groups, but its retention noticeably drops in the high-risk group and for smaller values of $\alpha$. MACI's log-product score has a clear motivation from the oracle formulation, is compatible with the conformal calibration framework, and empirically performs best overall in our experiments, so we use it as the default choice in the main text. Sum-based (log-sum) aggregation remains a promising alternative that could be explored further with more refined design in future work.

Table 7: Comparison with Conformity Score Variants (Power-Mean ($\lambda = 2$), Log-Sum, and MACI's log-product) on MedLFQA, across false-claim risk groups.

| | Target Coverage: 80% ($\alpha = 0.2$) | | | | | | Target Coverage: 90% ($\alpha = 0.1$) | | | | | | Target Coverage: 95% ($\alpha = 0.05$) | | | | | |
|---|---|---|---|---|---|---|---|---|---|---|---|---|---|---|---|---|---|---|
| | Power-Mean | | Log-Sum | | MACI | | Power-Mean | | Log-Sum | | MACI | | Power-Mean | | Log-Sum | | MACI | |
| Group | Cov. | Ret. | Cov. | Ret. | Cov. | Ret. | Cov. | Ret. | Cov. | Ret. | Cov. | Ret. | Cov. | Ret. | Cov. | Ret. | Cov. | Ret. |
| **MedLFQA: False-Claim Risk** | | | | | | | | | | | | | | | | | | |
| Low | 0.81 | 0.67 | 0.81 | 0.75 | 0.79 | **0.78** | 0.91 | 0.42 | 0.90 | **0.52** | 0.89 | **0.52** | 0.96 | 0.29 | 0.95 | 0.33 | 0.95 | **0.37** |
| Medium | 0.80 | 0.66 | 0.82 | **0.70** | 0.79 | **0.70** | 0.91 | 0.45 | 0.90 | **0.46** | 0.91 | **0.46** | 0.95 | **0.31** | 0.95 | **0.31** | 0.95 | **0.31** |
| High | 0.81 | 0.54 | 0.80 | 0.60 | 0.80 | **0.64** | 0.91 | 0.35 | 0.91 | 0.36 | 0.89 | **0.41** | 0.96 | 0.23 | 0.95 | 0.24 | 0.95 | **0.26** |

**Modeling Joint Probability.** Our primary experimental setting decomposes each model response into independent atomic claims and then estimates a factuality-score for each claim using only the $\{\text{Prompt}, c_i\}$ pair. This design follows the oracle formulation and enables conformal analysis at the claim level, implicitly adopting an independence assumption between claims. In realistic LLM outputs, however, claims may exhibit correlations, redundancy, or logical dependencies, so this assumption does not perfectly match the underlying generation process.

In our framework, factuality checking is delegated to $M$ LLMs used as verifiers, rather than being resolved within a single decoding of the generative model. Under this perspective, an important consideration is whether a claim-wise modeling strategy remains effective compared to a joint modeling approach that explicitly exposes the full Claim Set. To investigate this, we complement the per-claim setting with a joint modeling experiment in which the verifier receives the entire Claim Set at once and outputs conditional scores $\{p(c_i \mid C)\}_{i=1}^{N_i}$. Concretely, we provide $\{\text{Prompt}, \{c_1, \ldots, c_{N_i}\}\}$ as input so that the verifier can, in principle, exploit interactions among claims when assigning factuality-scores.

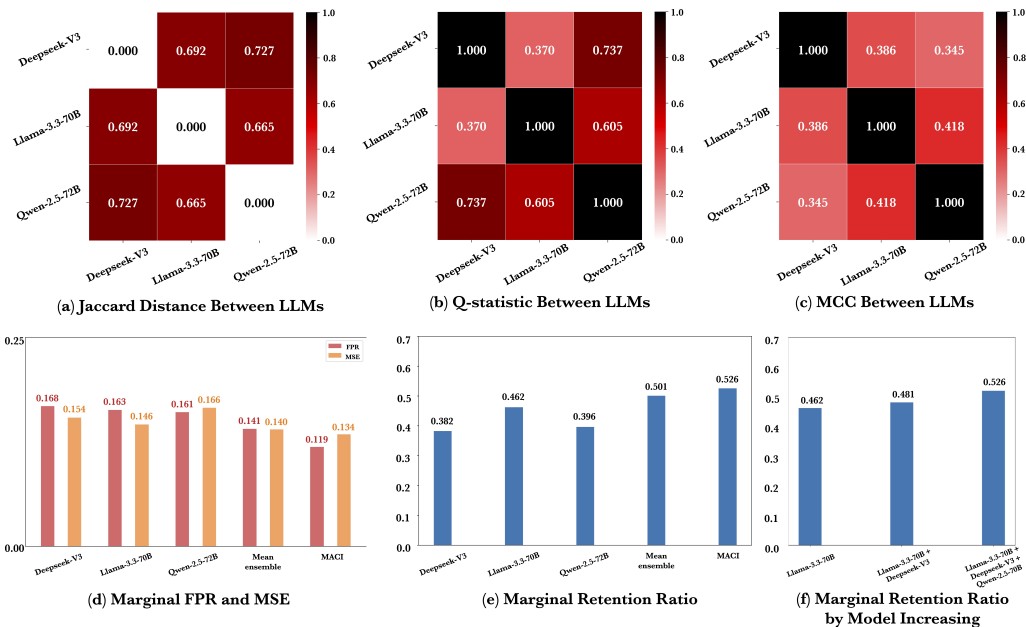

Figure 7: (a)-(c) display diversity diagnostics including Jaccard distance, Q-statistic, and MCC; (d)-(e) show the performance metrics of MACI; and (f) illustrates the retention trend by ensemble size. (a), (b), and (c) reveal high disagreement and low correlation between LLMs' predictions on false-claims, indicating diverse detection patterns that strongly support the necessity of using an ensemble. (d) demonstrates the sequential improvement in FPR and MSE from a single-LLM and arithmetic mean ensemble to our proposed MACI, while (e) shows that as these error rates improve, the retention ratio also increases. (f) demonstrates that as the number of models in the ensemble increases ($k = 1, 2, 3$) with the optimal combination selected, the retention ratio consistently improves, validating the efficiency of scaling the ensemble.

Table 8 shows that the per-claim setting and the joint setting yield similar coverage and retention profiles. This suggests that, even when the full Claim Set is available, the verifier LLM can effectively treat each claim as a comparatively independent decision unit, and that claim-wise modeling is a practically viable choice from the verifier's perspective. In other words, the independence assumption used for our oracle-based analysis serves both as a convenient theoretical simplification and as an approximation that does not substantially degrade performance when compared to a more joint modeling strategy.

Table 8: Comparison between MACI and MACI-Joint. MACI-Joint scores claims jointly given the whole claim set, while MACI scores each claim independently.

| | Target Coverage: 80% ($\alpha = 0.2$) | | | | Target Coverage: 90% ($\alpha = 0.1$) | | | | Target Coverage: 95% ($\alpha = 0.05$) | | | |
|---|---|---|---|---|---|---|---|---|---|---|---|---|
| | MACI | | MACI-Joint | | MACI | | MACI-Joint | | MACI | | MACI-Joint | |
| Group | Cov. | Ret. | Cov. | Ret. | Cov. | Ret. | Cov. | Ret. | Cov. | Ret. | Cov. | Ret. |
| **MedLFQA: False-Claim Risk** | | | | | | | | | | | | |
| Low | 0.79 | **0.78** | 0.80 | 0.75 | 0.89 | **0.52** | 0.91 | **0.52** | 0.95 | **0.37** | 0.95 | 0.36 |
| Medium | 0.79 | 0.70 | 0.80 | **0.74** | 0.91 | 0.46 | 0.90 | **0.47** | 0.95 | 0.31 | 0.95 | **0.34** |
| High | 0.80 | **0.64** | 0.80 | 0.60 | 0.89 | **0.41** | 0.90 | 0.38 | 0.95 | **0.26** | 0.95 | 0.25 |

# F    STATEMENT ON THE USE OF LARGE LANGUAGE MODELS

We used an LLM for minor editing and scripting automation only; core ideas, experiments, and analyses were conducted by the authors.

