# OpenReview forum: "Multi-LLM Adaptive Conformal Inference for Reliable LLM Response"
_ICLR.cc/2026/Conference — ICLR 2026 Poster_

### Official Review · Reviewer_fF7o · 2025-10-19

**Soundness:** 3
**Presentation:** 3
**Contribution:** 3
**Rating:** 6
**Confidence:** 3

**Summary:**

The paper proposes a conformal inference for the distribution-free uncertainty quantification. Overall, the paper is well-written and well-positioned. Two major contributions come to my attention: 1) the conformal inference is used to select the factual LLM-generated claims with group-conditional guarantees; 2) multi-LLM ensemble is used to get a preciser score estimator $\hat{p}$ with higher retention rate. The proposed method is evaluated on three datasets for empirical benchmark.

**Strengths:**

Typically for conformal inference, the coverage guarantee is expected. The authors have shown the (marginal and group-conditional) coverage guarantees in Theorem 1 and Theorem 2. Also, authors provided Theorem 3, in which the improved accuracy in the estimated score $\hat{p}$ will lead to more efficient retention. The motivation to multi-LLM ensemble is intuitive on top of Theorem 3. The flow of the paper is clear and reasonable. The empirical results align with the expectations.

**Weaknesses:**

A shortlist of weaknesses that might be worthy to address

1. The review of the related works on conformal inference seems not sufficient in my view. The authors do not mention too many papers addressing the group-conditional coverage. A incomplete list is
* Conditional validity of inductive conformal predictors
* [Neurips 2024] Class-conditional conformal prediction with many classes
* [ICML 2025] Bridging Fairness and Efficiency in Conformal Inference

2. Second, I am concerned about the conditional independence assumption in Definition 1. Since the claims are generated by LLM from **the same one** prompt, it does not intuitive to assume the factuality of each claim is independent. Did authors try to model the probability of claims being true jointly given one prompt? I understand it can break the way to obtain the calibration threshold for the conformal inference. But it might be a good sensitivity check, that the modeling the joint probability of the claims being true is similar to modeling them independently.

3. Third, I am confused by the dataset used in Multi-LLM ensemble. In specific, what is the M here. From my understanding, the prompt is repeated M times. But how can you make sure that the claim $c$ is the same in the repeated prompts to get the ensemble scores?
Do you consider Some of the atomic claims are considered the same?

**Questions:**

None

---

> ### Author Response · Authors · 2025-11-21
> **Response to Reviewer fF7o**
>
> ### Answer for Weakness 1
>
> Thank you for your thoughtful advice. **We have substantially revised the Related Work section** to significantly strengthen the discussion on group-conditional coverage. First, we start with BCI [6], which provides the existing foundational framework, and clearly state that its limitation is confined to marginal coverage, which only guarantees error rates over the entire distribution. Next, we mention the necessity and emergence of group-conditional coverage, systematically organizing the latest methodologies for strengthening group-conditional guarantees (such as group clustering) and the research flow toward multivalid coverage guarantees.
>
> Specifically, we introduce methods proposed by Ding et al. [5] and Gao et al. [8] that utilize class/group clustering and surrogate information, alongside research satisfying calibration simultaneously for multiple computable subgroups via multicalibration, and the multivalid coverage concept presented by Jung et al. [1]'s Batch Multivalid Conformal Prediction. These approaches emphasize satisfying class-conditional or group-conditional coverage for numerous classes or groups while analyzing the trade-off with prediction set size (efficiency) and proposing directions for improvement. Additionally, it further explains the context of applying group-conditional guarantees to LLM factuality assurance, including Liu & Wu et al. [2]'s multi-group uncertainty quantification and Detommaso et al. [4]'s multicalibration-based LLM confidence calibration research. In the context of this group-conditional coverage, the objective of MACI is clearly established as simultaneously satisfying group-conditional guarantees and a practically achievable high retention ratio. MACI's positioning is thus clarified by noting that its strongest baseline CCI [7] shares the same fundamental goal.
>
> We go beyond merely discussing these methods in the Related Work section; in Appendix E, we actually implement these methodologies and perform comparative analysis against MACI. We implement Ding/Gao-style group/class clustering-based group-conditional methods and Jung et al.'s multivalid-based (MVCI) methods tailored for false-claim filtering settings, quantitatively comparing them with MACI in terms of coverage and retention. The comparison demonstrates that while existing multivalid methods provide strong guarantees, they tend to yield conservative predictions, significantly reducing retention. In contrast, MACI achieves a practically usable retention ratio under realistic group definitions, sacrificing strict group-conditional coverage.

---

> ### Author Response · Authors · 2025-11-21
> **Response to Reviewer fF7o**
>
> ### Answer for Weakness 2
>
> Thank you for raising this important point. In Section 4.2 (Adaptive Conformal Inference for False-Claim Filtering), we now explicitly distinguish between the roles of validity and efficiency. The conditional independence assumption in Definition 1 is not required for conformal validity. The group-conditional coverage guarantee follows solely from exchangeability, so validity continues to hold even when factuality labels generated from the same prompt exhibit arbitrary dependence.
>
> Conditional independence is used only in our efficiency analysis, where the goal is to define a tractable and interpretable oracle benchmark. Its role is analogous to the use of conditional independence in the naive Bayes classifier. In naive Bayes, the assumption is not intended to be realistic, but it enables a factorization of the joint likelihood that would otherwise be infeasible to estimate, yielding a simple and well-structured scoring rule that often performs surprisingly close to the true Bayes classifier. The value of the assumption lies in the clarity it provides rather than in strict realism.
>
> Our use of conditional independence serves the same purpose. It allows the oracle joint factuality
>
> $$
> \mathbb{P}\big(Y_1=1, \ldots, Y_m=1 \mid P, C\big)
> $$
>
> to factorize into a product of marginal oracle scores, making the multiplicative rule the natural analogue of the oracle decision. Without this factorization, the oracle benchmark becomes unidentifiable, and the efficiency analysis loses interpretability. The assumption therefore clarifies the structural conditions under which the multiplicative score is optimal, while remaining unnecessary for distribution-free coverage.
>
> Following your suggestion, we also examine a setting in which the probability of all claims being true is modeled jointly rather than marginally. The table below reports the results of an additional sensitivity experiment where the marginal oracle scores were replaced by a jointly modeled oracle that evaluates the entire claim set within each document at once. The joint-construction removes the conditional independence approximation and captures the full dependence structure among claims produced from the same prompt. The resulting filtering performance is highly similar to that obtained under the marginal formulation, indicating that the conclusions of our efficiency analysis are stable even when the oracle is constructed from the joint distribution rather than from independent components.
>
> |                   |         | **$\alpha = 0.2$** |               | **$\alpha = 0.1$** |               | **$\alpha = 0.05$** |               |
> | :---------------: | :-----: | :---------------------------: | :-----------: | :----------------------------: | :-----------: | :-----------------------------: | :-----------: |
> | **False-Claim Risk**         | **Metric** | **MACI**                   | **MACI-Joint** | **MACI**                     | **MACI-Joint** | **MACI**                      | **MACI-Joint** |
> | **Low**           | **Cov.** | 0.79                        | 0.80          | 0.89                          | 0.91          | 0.95                           | 0.95          |
> |                   | **Ret.** | **0.78**                    | 0.75          | **0.52**                      | **0.52**      | **0.37**                       | 0.36          |
> | **Medium**        | **Cov.** | 0.79                        | 0.80          | 0.91                          | 0.90          | 0.95                           | 0.95          |
> |                   | **Ret.** | 0.70                        | **0.74**      | 0.46                          | **0.47**      | 0.31                           | **0.34**      |
> | **High**          | **Cov.** | 0.80                        | 0.80          | 0.89                          | 0.90          | 0.95                           | 0.95          |
> |                   | **Ret.** | **0.64**                    | 0.60          | **0.41**                      | 0.38          | **0.26**                       | 0.25          |
>
>
>
>
> To improve transparency, Section 4.2 has been reorganized so that conditional independence is used only in the efficiency analysis, and we clearly separate the distribution-free validity argument from the structural assumptions used to analyze efficiency and oracle optimality.

---

> ### Author Response · Authors · 2025-11-21
> **Response to Reviewer fF7o**
>
> ### Answer for Weakness 3
>
> Thank you for your important question. The main points are (1) what $M$ means in the Multi-LLM ensemble, and (2) how we compute ensemble scores consistently for the same claim.
>
> In our ensemble, $M$ is the number of distinct LLMs used as factuality verifiers. We use QA-based datasets that are pre-constructed for the false-claim filtering task. Each sample is given in the form
> $$
> \(\text{Prompt}, \text{Response}, \text{Claim Set}, \text{Ground Truth}\).
> $$
> Here, the Claim Set is a collection of atomic claims, obtained by decomposing the response into atomic units at dataset construction time. For each atomic claim, the corresponding ground-truth label is also fixed in the dataset. We use this Claim Set as is, and there is not a step where we repeatedly generate responses for the same prompt or re-extract claims for the purpose of ensembling.
>
> The Multi-LLM ensemble proceeds as follows. We first select $M$ different LLMs as factuality verifiers. Then, for each dataset sample $\{(\text{Prompt}, \text{Response}, \text{Claim Set})\}$, we keep $\text{Prompt}$ and the $\text{Claim Set}$ fixed and feed each atomic claim to all verifier LLMs. Concretely, for each atomic claim $c$ in the $\text{Claim Set}$, every verifier LLM receives the same $\{(\text{Prompt}, c)\}$ pair as input and outputs a factuality-score (interpreted as the probability that $c$ is factually correct). This yields an $M$-dimensional score vector for each claim,
> $$
> \bigl(p_1(c), p_2(c), \dots, p_M(c)\bigr),
> $$
> and MACI aggregates this vector into a single final factuality-score via its ensemble procedure. An example of prompt template used for factuality verification (not the $\text{Prompt}$ above) is described in detail in Appendix D.
>
> In conclusion, the $\text{Response}$ and $\text{Claim Set}$ is defined once at the dataset-construction stage as a set of atomic claims, and we simply collect verbalized factuality-scores for each fixed claim from multiple LLMs and then ensemble them.
>
> To make this clearer, we have added a detailed description at the beginning of Section 5 (Empirical Results).

---

### Official Review · Reviewer_p4aF · 2025-10-28

**Soundness:** 3
**Presentation:** 3
**Contribution:** 2
**Rating:** 6
**Confidence:** 3

**Summary:**

The paper tackles reliable factuality filtering for LLM outputs using conformal inference (CI). Prior CI approaches either (i) apply a single global threshold (achieving marginal coverage but aggressively filtering, hence low retention) or (ii) learn sample-wise thresholds with adaptive error rates which improves retention but weakening fixed-level guarantees. The authors propose MACI, which (1) reformulates factuality filtering via a multiplicative conformity score (cumulative probability product across claims), (2) introduces group-conditional calibration to retain finite-sample, distribution-free guarantees within predefined subgroups, and (3) increases retention by improving factuality-score quality through a multi-LLM ensemble whose weights are optimized. They also give retention analysis in this CI setting, linking deviations from an oracle factuality-score to true-claim preservation, which motivates the ensemble design. Empirically, MACI maintains user-specified coverage while substantially improving retention and reducing time cost relative to BCI and CCI baselines; an anonymized repository is provided.

**Strengths:**

1- Casting filtering as a cumulative probability product of claim-level factuality scores is natural yet non-obvious; it aligns the conformity score with joint “no-false-claims” events and helps reconcile coverage with retention.

2- The paper provides a retention analysis linking oracle-estimator deviations to preserved true claims, thereby justifying the multi-LLM ensemble design rather than treating it as a heuristic.

3- Group-conditional calibration preserves finite-sample guarantees at user-specified risk levels while avoiding the heavy conservatism of global thresholds; it also sidesteps adaptive-α issues in high-stakes settings.

4- The method achieves higher retention at target coverage and shows favorable time-cost due to single-pass scoring and simpler calibration vs. conditional boosting/sampling methods.

**Weaknesses:**

1- The method relies on exchangeability/i.i.d. between calibration and test; real deployments may face distribution shift (domain, style, model temperature). The paper would benefit from experiments that explicitly stress test group-conditional coverage under covariate shift (e.g., density-ratio buckets, temporal drift).

2- Guarantees are group-conditional, so the definition of groups meaningfully affects validity/retention. More analysis on group granularity vs. sample size trade-offs (and automatic group discovery) would strengthen the story.

3- Retention gains hinge on factuality-score quality and thus on ensemble diversity and calibration. The paper could add ablations on ensemble composition, weighting stability, and computational budget vs. benefit.

4- While the multiplicative score is well-motivated, exploring alternatives (e.g., log-sum, powered products) and reporting sensitivity would clarify robustness to mis-calibrated claim probabilities. (This is partly implied by the theory but not fully evaluated.)

5- Many high-stakes domains require retrieval/attribution. It would help to discuss compatibility with retrieval-augmented pipelines and very long contexts where claim segmentation and score independence may be more brittle. (Background mentions database-heavy approaches; MACI’s stance could be clarified.)

**Questions:**

1- How does MACI’s group-conditional coverage behave under covariate shift (prompt style, domain drift, temperature changes)? Could the authors report coverage/retention under controlled density-ratio bins or apply lightweight split-conformal covariate-shift corrections?

2- What guidance can you offer for choosing group granularity 𝐾 given a fixed calibration budget? Any results on automatic grouping (e.g., clustering claim/prompt features) vs. fixed datasets’ categories?

3- Please include diversity diagnostics (pairwise disagreement, Q-statistic) and an efficiency-quality frontier varying the number/type of models and weighting schemes. How sensitive is MACI to over-reliance on a single strong model?

4- Did you compare the multiplicative conformity score to additive/log-sum or power-mean variants? A small ablation would help establish that the product is not only principled but also empirically dominant.

5- Table 3 is helpful; could you add wall-clock numbers at larger evaluation scales and with streaming recalibration when groups shift or new domains appear?

---

> ### Author Response · Authors · 2025-11-21
> **Response to Reviewer p4aF**
>
> ### Answer for Question 1
>
> Thank you for raising the question about MACI’s behavior under covariate shift. We have added a new subsection Section 5.3 (MACI under Covariate Shift) in the main text and a more detailed description in Appendix E, where we construct an explicit covariate shift and evaluate MACI (and a density-ratio–corrected variant, MACI-DRE) in that setting.
>
> To create a covariate shift on MedLFQA, we explicitly model **difficulty**. For each sample \(x = (P, R, C)\), we apply Manakul et al. [15]'s SelfCheck score at the claim level and define
>
> $$
> s(x) = \frac{1}{m(x)} \sum_{j=1}^{m(x)} \text{SelfCheck}_j,
> $$
>
> where $m(x)$ is the number of atomic claims in the response. Larger $s(x)$ corresponds to easier, more reliable responses; smaller $s(x)$ corresponds to harder, more hallucination-prone responses. We sort all samples by $s(x)$ and then:
>
> - use the higher part of this ordering (easier prompts/responses) as the calibration pool,
> - use the lower part (harder prompts/responses) as the test pool.
>
> This yields a clear covariate shift in the difficulty-related features while keeping the annotation process fixed.
>
> We then introduce a lightweight density-ratio–corrected version of MACI, based on the density estimation of Tibshirani et al. [10], which we call **MACI-DRE**. We estimate a density ratio
> $$
> r(x) = \frac{p^{(t)}_X(x)}{p^{(s)}_X(x)}
> $$
> from simple, deployment-feasible features (summaries of $(s(x)$, its standard deviation, and prompt/response lengths), and use this to construct a **density-ratio–weighted calibration set** via importance resampling. We then run the original MACI algorithm on this resampled calibration set, without changing any internal components of MACI (ensemble weight learning, subgroup-wise threshold estimation).
>
> Finally, we compare MACI and MACI-DRE under this difficulty-based covariate shift using the same false-claim–risk group definitions. We report group-conditional coverage and retention for both methods; the results are summarized in the table below (grouped by false-claim risk and target coverage levels):
>
> | | | **$\alpha = 0.2$** | | **$\alpha = 0.1$** | | **$\alpha = 0.05$** | |
> | :---: | :---: | :---: | :---: | :---: | :---: | :---: | :---: |
> | **False-Claim Risk** | **Metric** | **MACI** | **MACI-DRE** | **MACI** | **MACI-DRE** | **MACI** | **MACI-DRE** |
> | **Low** | **Cov.** | 0.68 | 0.76 | 0.85 | 0.93 | 0.94 | 0.95 |
> | | **Ret.** | 0.83 | 0.72 | 0.57 | 0.35 | 0.34 | 0.29 |
> | **Medium** | **Cov.** | 0.65 | 0.84 | 0.82 | 0.94 | 0.87 | 0.96 |
> | | **Ret.** | 0.77 | 0.56 | 0.55 | 0.27 | 0.42 | 0.19 |
> | **High** | **Cov.** | 0.77 | 0.75 | 0.88 | 0.89 | 0.92 | 0.93 |
> | | **Ret.** | 0.62 | 0.67 | 0.39 | 0.38 | 0.27 | 0.28 |
>
> Empirically, we observe that under this difficulty-based covariate shift, MACI’s group-conditional coverage can fall below the nominal targets, especially for low- and medium-risk groups at tighter coverage levels. In contrast, MACI-DRE consistently brings the group-conditional coverage much closer to the desired levels across all false-claim–risk groups and all values of \(\alpha\), indicating that the density-ratio–based correction effectively compensates for the covariate shift in this setting. Detailed descriptions of this construction and additional results are provided in Section 5.3 and Appendix E of the revised manuscript.

---

> ### Author Response · Authors · 2025-11-21
> **Response to Reviewer p4aF**
>
> ### Answer for Question 2
>
> Thank you for your constructive question. Your question concerns (i) how to choose the number of groups $K$ given a fixed calibration budget, and (ii) whether one can use automatic grouping (e.g., clustering based on claim/prompt features) instead of pre-defined dataset categories.
>
> First, the base version of MACI in our paper follows a Mondrian-style group-conditional conformal scheme: for each user-defined group $g \in G$, we learn a separate threshold. When the calibration sample size $n_g$ for a given group is small, the empirical $(1-\alpha)$-quantile $\hat{q}_{1-\alpha,g}$ has high variance. This creates a natural trade-off: we retain strict group-conditional guarantees, but group-wise coverage and retention can become unstable when there are many small groups.
>
> To mitigate this issue, we construct MACI-Cluster by combining MACI with the group clustering idea of Gao et al. [8], and we report detailed experiments in Appendix E. Concretely:
>
> We first define a finer-grained partition of groups (e.g., 24 groups) based on the false-claim risk score, deliberately creating a scenario with many small groups (less than 50 calibration samples). We then compare the conformity-score distributions of these groups, cluster those with similar distributions together, and share a single calibrated threshold at the cluster level. As shown in Figure 5 and Appendix E, while the original MACI achieves group-wise average coverage closer to the target level, it exhibits higher variance across groups and runs. In contrast, MACI-Cluster substantially reduces this variance, offering much more stable coverage and thresholds in settings with many groups and small sample sizes, albeit with a small bias relative to the target level.
>
> Based on these observations, a practical guideline under a fixed calibration budget $n$ is:
>
> 1. **Start from domain-meaningful “base groups.”**
>    For example, groups can be defined using false-claim risk, domain (medical / Wikipedia / expert), prompt style, or other semantically meaningful categories.
>
> 2. **Choose $K$ so that each group has a sufficient calibration sample size $n_g$.**
>    In our experiments, we observe that when each group has more than $\sim 100$ calibration samples, the group-specific quantiles are reasonably stable. If $n/K$ is much smaller than this (so many groups are very small), it is usually better to use a clustered approach like MACI-Cluster rather than a pure Mondrian scheme.
>
> Regarding **automatic grouping vs. fixed categories**:
>
> In this paper, our automatic component is not direct clustering in the prompt/claim feature space. Instead, we:
>   1. First define fine-grained groups based on false-claim risk.
>   2. Then cluster these groups using their conformity-score distributions (following the spirit of Gao et al.) and share thresholds within each cluster.

---

> ### Author Response · Authors · 2025-11-21
> **Response to Reviewer p4aF**
>
> ### Answer for Question 4
>
> We appreciate your suggestion to compare the multiplicative conformity score with the additive, log-sum, and power-mean variants. Our choice of the multiplicative form is motivated by the oracle setting, where factuality probabilities factorize under conditional independence. This factorization makes the product score a natural and interpretable baseline.
>
> At the same time, we agree that an ablation study is valuable for assessing whether this choice also performs well empirically. Conformal validity is preserved under any symmetric transformation of the conformity score, so the primary effect of alternative transformations lies in efficiency. Following the reviewer’s recommendation, we added a sensitivity analysis comparing the multiplicative score with log-sum, power-mean, and other related variants, particularly in regimes where the underlying probabilities may be mis-calibrated.
>
> The additional experiments indicate that the multiplicative score remains competitive across all settings and often attains the highest retention, while coverage remains comparable across transformations. Representative results are summarized below.
>
> | | | | **$\alpha = 0.2$** | | | **$\alpha = 0.1$** | | | **$\alpha = 0.05$** | |
> | :---: | :---: | :---: | :---: | :---: | :---: | :---: | :---: | :---: | :---: | :---: |
> | **Risk** | **Metric** | **Power-Mean** | **Log-Sum** | **MACI** | **Power-Mean** | **Log-Sum** | **MACI** | **Power-Mean** | **Log-Sum** | **MACI** |
> | **Low** | **Cov.** | 0.81 | 0.81 | 0.79 | 0.91 | 0.90 | 0.89 | 0.96 | 0.95 | 0.95 |
> | | **Ret.** | 0.67 | 0.75 | **0.78** | 0.42 | **0.52** | **0.52** | 0.29 | 0.33 | **0.37** |
> | **Medium** | **Cov.** | 0.80 | 0.82 | 0.79 | 0.91 | 0.90 | 0.91 | 0.95 | 0.95 | 0.95 |
> | | **Ret.** | 0.66 | **0.70** | **0.70** | 0.45 | **0.46** | **0.46** | **0.31** | **0.31** | **0.31** |
> | **High** | **Cov.** | 0.81 | 0.80 | 0.80 | 0.91 | 0.91 | 0.89 | 0.96 | 0.95 | 0.95 |
> | | **Ret.** | 0.54 | 0.60 | **0.64** | 0.35 | 0.36 | **0.41** | 0.23 | 0.24 | **0.26** |
>
> We have included a short discussion of these results in Appendix E.

---

> ### Author Response · Authors · 2025-12-01
> **Response to Reviewer p4aF**
>
> ### Answer for Question 3
>
> Following the reviewer's suggestion, we analyze the diversity of the models composing the ensemble and the changes in performance relative to the number of models. We extend the analysis from Figure 3 in the main text to Figure 7 (in the Appendix) to provide a more detailed verification of the diverse false-claim detection patterns. Specifically, Figure 7(b) and (c) present the Q-statistic and MCC (Matthews Correlation Coefficient) diagnostics. Analyzing the correlations among the three models—Llama-3.3-70B, Qwen-2.5-72B, and DeepSeek-V3—regarding false-claims in the MedLFQA dataset reveals **a low positive correlation, with the Q-statistic (e.g., 0.370) and the MCC (e.g., 0.345).** This demonstrates that each model provides complementary information and that the quality of the factuality-score is substantially improved through ensembling, rather than relying solely on the single best-performing model. Furthermore, as shown in Figure 7(f), we analyze the change in the retention ratio as we increase the number of models in the ensemble from 1 to 3, selecting the optimal combination at each step. The experiment confirms that the marginal retention ratio monotonically increases to **0.462, 0.481, and 0.526** as the number of models increases. This indicates that MACI achieves higher retention through the ensemble effect rather than depending excessively on a specific strong model, suggesting that stable performance improvements are expected even when flexibly adjusting the number of models according to the computational budget.
>
>
> ### Answer for Question 5
>
> To validate efficiency in a deployed environment, we measure and compare the wall-clock time in a large-scale evaluation setting designed to simulate a real-world service scenario. The experiment is conducted using the WikiBio dataset, configured with 1,500 calibration samples and 500 test samples. We assume that factuality-scores for the calibration samples are pre-computed for both CCI and MACI, although it is worth noting that MACI is already significantly faster than CCI in the factuality-score generation phase, as shown in the same table. For the 500 test samples, we measure the time taken to generate factuality-scores via the OpenRouter API and perform the filtering. The results show that **CCI took 1643.91 seconds, whereas MACI took 598.98 seconds, completing the entire filtering pipeline**. Since this result includes the time for external API communication, the overall execution time would be further reduced if the models are run in a local environment.

---

### Official Review · Reviewer_eeqE · 2025-10-31

**Soundness:** 3
**Presentation:** 2
**Contribution:** 4
**Rating:** 4
**Confidence:** 4

**Summary:**

The authors propose a conformal method (MACI) for filtering LLM outputs. MACI leads to higher claim retention that prior work while still maintaining marginal/group covergage.

**Strengths:**

The paper is well-motivated and the proposed method is novel. The the empirical results (Table 1) are strong and clearly communicated.

**Weaknesses:**

In general, I think this is a good paper. However, I think the empirical section does not clearly communicate some details. For example, I believe completing replicating the experiments from just reading the paper (without contacting the authors or looking through their code) would be challenging. While I understand that the authors are limited in space, I also did not find it easy to parse some of these details out of the appendix.

I am open to improving my score once some of my clarifying questions and suggestions are addressed, but currently, I am not comfortable giving a higher rating.

**Questions:**

Questions (I apologize in advance if some of these questions were directly addressed in the main body):
1. How exactly do you produce estimates $\hat{p}$ / What is $\hat{p}$ your experiments? It wasn't clear to me if you were using base uncertainty scores like in prior work (verbalized confidence, frequency scores, model probabilities, P(True), etc.) or if the "black-box classifier" is also something you train for each LLM.
2. For the ensemble, do you mean that you are taking $\hat{p}$ from each LLM: Llama-3.3-70B-Instruct, Qwen-2.5-72B-Instruct, DeepSeek-V3?
3. Which model's output are you filtering? Or are you running these methods on outputs from all 3 and then taking an average in Table 1?

Suggestions:
1. I recommend citing a few other works related to group-conditional conformal prediction.
- [1] Jung, Christopher, et al. "Batch Multivalid Conformal Prediction." International Conference on Learning Representations.
- [2] Liu, Terrance, and Steven Wu. "Multi-group Uncertainty Quantification for Long-form Text Generation." The 41st Conference on Uncertainty in Artificial Intelligence.
- [3] Hébert-Johnson, Ursula, et al. "Multicalibration: Calibration for the (computationally-identifiable) masses." International Conference on Machine Learning. PMLR, 2018.
- [4] Detommaso, Gianluca, et al. "Multicalibration for Confidence Scoring in LLMs." International Conference on Machine Learning. PMLR, 2024.

    In contrast to Cherian et al. 2024 / Gibbs et al., 2024, [1] and [2] frame group-conditional conformal prediction more similarly to works on multicalibration (like [3], [4]), and they also typically consider more groups than in your work or that of Cherian et al. 2024. Potentially, you could also compare to the methods evaluated in [2] (group-conditional versions split conformal and conformal quantile regression) to further bolster your experimental results since at the end of the day, the goal of those methods are also to achieve better group coverage and retention rates. However, I think just mentioning these lines of works would be sufficient.
2. I don't see the value of comparing to sampling-based methods (SelfCheck, FSC) since as you stated, the goal of those methods are not to achieve some target coverage. I think the point you're making could also be made with any conformal method like just using BCI, so the comparison doesn't highlight to me anything about MACI. Hearing your thoughts on this would be helpful. In my opinion, you could remove these experiments without hurting the paper at all, and it would leave more room to explain other details in the main body.

    Also, since these methods are in some sense, proposing better options for $\hat{p}$, perhaps a more sensible comparison might have been to see what happens if you use these sampling methods as the base uncertainty score $\hat{p}$ (instead of the "frequency score" that prior work often use) for BCI, CCI, multi-valid split conformal, etc. However, maybe that would make sense if BCI/CCI perform better using SelfCheck/FSC compared to frequency scores and that MACI still performs the best.
3. Perhaps Figure 3b and c should be bar charts instead of line graphs.
4. I think 5.2 and 5.3 could be moved to the appendix so that some other experimental details from the appendix can be put into the main body.

---

> ### Author Response · Authors · 2025-11-21
> **Response to Reviewer eeqE**
>
> ### Answer for Q1,Q2,Q3
>
> Thank you for your thoughtful question. We utilize the datasets modified from existing QA datasets to suit the false-claim filtering task, thereby validating MACI's excellence. The structure of one data sample follows the $\text{\{Prompt, Response, Claim Set, Ground Truth\}}$, which is a common validation environment used in existing studies such as Hashimoto et al. (BCI) [6] and Cherian et al. (CCI) [7]. MACI leverages this well-structured data to validate its filtering performance.
>
> **Factuality-score Calculation Process (Answer for Q1 & Q2)**
>
> MACI calculates the final factuality-score by ensembling outputs from various models. The factuality-score calculation can be simplified into the following two steps:
>
> *Step 1: Obtaining Verbalized Factuality-Scores*
>
> Each of the $M$ LLMs participating in the ensemble receives a specially designed instruction prompt, such as: “You are the world's best factuality verification expert & logician. Infer the probability that the given {Prompt, claim} is true as a value between 0 and 1.” Each model outputs a verbalized factuality-score for every claim in the claim set according to this prompt.
>
> *Step 2, Final Factuality-Score*
>
> The verbalized factuality-scores obtained from these $M$ models undergo an optimization ensemble process according to MACI's algorithm (Algorithm 2 in the main text) and are integrated into a single final factuality score.
>
> **Filtering Target (Answer for Q3)**
>
> MACI is a general filter not limited to responses from a specific LLM. We utilize pre-existing datasets, which already contain LLM responses to $\text{Prompt}$. Therefore, MACI filters responses from the LLMs used to create these datasets. Responses in each dataset were generated by various LLMs, such as gpt-4 and gpt-3.5-turbo.
>
> To clarify this point, we have supplemented the relevant information at the beginning of Section 5: Empirical Results in the main text.

---

> ### Author Response · Authors · 2025-11-21
> **Response to Reviewer eeqE**
>
> ### Answer for Suggestion 1
>
> We appreciate the insightful comments outlining the research trajectory of the Multivalid Conformal Prediction family.
>
> Jung et al. [1]'s Batch Multivalid Conformal Prediction presents a robust framework that simultaneously guarantees both group-conditional coverage for multiple subgroups and threshold-level-conditional coverage for the threshold levels generated by the model. However, MACI's primary goal is to achieve a practical retention ratio while maintaining group-conditional coverage for meaningful groups in the false-claim filtering task for LLM-generated responses. In this problem setting, we believe Cherian et al. (CCI) [7] , which shares the same false-claim filtering environment and evaluation protocol, serves as the strongest baseline directly corresponding to MACI's objective. Based on this feedback, we have substantially enhanced the Related Work section to strengthen the narrative, clearly contrasting the objectives and positioning of our work with Multivalid and multicalibration-related studies.
>
> Furthermore, to solidify the rationale for the enhanced narrative, we implemented the MultiValid Conformal Inference (MVCI) method, which is a modification of Jung et al.'s BatchMVP tailored for the false-claim filtering task, and conducted additional experiments directly comparing coverage and retention between MACI and MVCI at the group-conditional level. The table below demonstrates that while MVCI calculates a highly conservative threshold to simultaneously achieve group-conditional and threshold-level-conditional guarantees, MACI maintains the same level of group-conditional coverage while exhibiting significantly higher retention. This result can be interpreted as stemming from different design points: MVCI pursues stronger multivalid coverage guarantees, while MACI aims for practical false-claim filtering that simultaneously achieves group-conditional level guarantees and a feasible retention ratio. We detail these additional experimental results in Appendix E.
>
> | | | **$\alpha = 0.2$** | | **$\alpha = 0.1$** | | **$\alpha = 0.05$** | |
> | :---: | :---: | :---: | :---: | :---: | :---: | :---: | :---: |
> | **WikiBio & False-Claim Risk** | **Metric** | **MVCI** | **MACI** | **MVCI** | **MACI** | **MVCI** | **MACI** |
> | **Low** | **Cov.** | 0.80 | 0.82 | 0.89 | 0.90 | 0.95 | 0.94 |
> | | **Ret.** | 0.11 | **0.40** | 0.03 | **0.23** | 0.02 | **0.17** |
> | **Medium** | **Cov.** | 0.81 | 0.81 | 0.91 | 0.90 | 0.96 | 0.95 |
> | | **Ret.** | 0.08 | **0.42** | 0.02 | **0.25** | 0.01 | **0.12** |
> | **High** | **Cov.** | 0.81 | 0.81 | 0.90 | 0.90 | 0.95 | 0.96 |
> | | **Ret.** | 0.03 | **0.45** | 0.01 | **0.28** | 0.01 | **0.09** |

---

> ### Author Response · Authors · 2025-11-21
> **Response to Reviewer eeqE**
>
> ### Answer for Suggestion 2
>
> **Response to Sampling-based Methods**
>
> Thank you for your advice and for understanding our concern about page limits. As mentioned in the main text, since sampling-based methods (SelfCheck, FSC) do not provide statistical guarantees, it is reasonable not to perform comparisons from a statistical guarantee perspective. However, sampling-based methods are a methodology that enhances the factuality of LLM responses applicable to black-box LLMs. I believe this represents a research stream sharing a similar goal to MACI from the broader perspective of developing a false-claim filter applicable to black-box LLMs. Therefore, I still consider analyzing the empirical coverage and retention of sampling-based methodologies meaningful. However, as you suggested, to allocate the main text space to detailing the experimental section, we have moved this part to Appendix E.
>
> **Response to Proposed Additional Experiments**
>
> BCI and CCI use frequency scores as their factuality-scores in their methodologies, and these frequency scores are generated through the SelfCheck method, which is a sampling-based approach. Thus, BCI and CCI already employ factuality-scores derived from sampling-based methods. Nevertheless, the significance of the additional experiment you proposed lies in verifying that MACI's oracle-motivated adaptive conformal inference structure outperforms the baseline structure, even if the quality of the factuality-score (uncertainty score) is the same across BCI, CCI, and MACI. We consider this a good idea. Therefore, we fix MACI's factuality-score to a frequency score. That is, we compare the coverage and retention of MACI with BCI and CCI, excluding the Multi-LLM ensemble component. The table below shows that MACI achieves the highest retention even under the same unified factuality-score. This demonstrates that MACI's oracle-motivated adaptive conformal inference structure itself is superior to the baselines. We have added these experimental results to Appendix E.
>
> | | | | **$\alpha = 0.2$** | | | **$\alpha = 0.1$** | | | **$\alpha = 0.05$** | |
> | :---: | :---: | :---: | :---: | :---: | :---: | :---: | :---: | :---: | :---: | :---: |
> | **False-Claim Risk** | **Metric** | **BCI** | **CCI** | **MACI** | **BCI** | **CCI** | **MACI** | **BCI** | **CCI** | **MACI** |
> | **Low** | **Cov.** | 0.84 | 0.83 | 0.82 | 0.94 | 0.91 | 0.91 | 0.97 | 0.95 | 0.96 |
> | | **Ret.** | 0.07 | 0.68 | **0.70** | 0.03 | 0.41 | **0.49** | 0.01 | 0.28 | **0.32** |
> | **Medium** | **Cov.** | 0.83 | 0.81 | 0.82 | 0.89 | 0.90 | 0.91 | 0.94 | 0.95 | 0.96 |
> | | **Ret.** | 0.06 | 0.66 | **0.71** | 0.03 | 0.39 | **0.46** | 0.01 | 0.25 | **0.30** |
> | **High** | **Cov.** | 0.73 | 0.78 | 0.81 | 0.88 | 0.89 | 0.90 | 0.94 | 0.94 | 0.95 |
> | | **Ret.** | 0.06 | 0.43 | **0.58** | 0.01 | 0.22 | **0.36** | 0.01 | 0.12 | **0.24** |

---

> ### Author Response · Authors · 2025-11-21
> **Response to Reviewer eeqE**
>
> ### Answer for Suggestion 3,4
>
> Thank you for the suggestion. We have revised Figure 3b and 3c into clearer line plots to improve readability. We decided to keep Sections 5.2 and 5.3 in the main body because they report the time-cost and practical efficiency of MACI, which are crucial for real-world, high-throughput LLM filtering systems. In fact, another reviewer explicitly emphasized the importance of computational cost, so we think these results belong in the main text. Instead, the experimental setup will be clearly detailed at the beginning of Section 5.

---

> > ### Comment · Reviewer_eeqE · 2025-11-25
> >
> > Thank you for addressing my concerns, I will raise my score.

---

### Official Review · Reviewer_4aeR · 2025-11-01

**Soundness:** 3
**Presentation:** 4
**Contribution:** 3
**Rating:** 8
**Confidence:** 4

**Summary:**

The paper proposes a subgroup-aware selective prediction procedure that learns a filter and threshold per subgroup to control FPR while maximizing retention and attains valid subgroup-conditional coverage. An ensemble over base filters stabilizes performance, and the method reports strong empirical FPR control with competitive efficiency. I appreciate the clear motivation and the practical design choices (e.g., monotonicity of FPR in the tuning parameter, which enables efficient search). My main concerns are (i) statistical stability in underrepresented groups when learning per-group thresholds/filters, (ii) the strength and usefulness of Assumptions 1–2 (Theorem 3 hinges on them), and (iii) limited discussion of PAC-style guarantees despite the monotone FPR structure. I also encourage a more robust experimental protocol (more than 30 trials). More discussion on retention-aware, adaptive-α strategy would be good.

**Strengths:**

* Clear problem framing at the intersection of fairness (group FPR control) and efficiency (retention), with a practical ensemble that helps smooth variance across base filters.

* Good use of the monotonic relation between the parameter and FPR, which opens the door to simple line or binary searches and possible RCPS-style risk control.

* Careful numerical studies and sensible metrics; the approach feels implementable.

**Weaknesses:**

* Learning a separate filter and threshold per subgroup can be statistically brittle when the sample size is small, such as in underrepresented subgroups; estimates of group-level score quantiles and FPR could swing widely, harming both fairness and efficiency. Consider clustering subgroups with similar score distributions (cf. Gao et al., 2025 or Ding et al, 2023) or hierarchical / shrinkage pooling of quantiles.

* As written, C appears tunable or possibly loose enough that the assumption risks being vacuous, i.e., I could take C large enough to satisfy the assumption. More clarification on whether C is pop-determined, empirically verifiable, or enters the finite-sample bound in a way that tightness matters.

* Assumption 1 and 2 seem potentially strong for minority groups; if score density at the operating quantile is small, variance explodes, and makes the bound useless. More discussion on failure modes when density at decision boundary is low is needed.

* Because FPR is monotone in the parameter, RCPS techniques suggest that PAC coverage bounds could be obtained.

* 30 trials used in Table 1 seems small. How sensitive are results to group size imbalance?

**Questions:**

1. If test and calibration data distributions differ or if the filter is learned on the same data as thresholding without sample splitting, nominal control may not hold. It would be good to see clear explicit cross-fitting across subgroups (which could be unstable without clustering those with similar conformal scores as suggested) or an accounting for potential dataset shift, e.g., covariate shift with bounds on the density ratio.

2. If C upper bounds either the change in FPR as a function of the parameter near the operating point or the inverse density, then taking it to be arbitrarily large makes the bound true but uninformative. Can C be tied to an observable quantity, e.g., plug-in estimator of f_S at the quantile with high-prob bounds? Or can it be acknowledged that small density at the threshold leads to large constants and loose rates; can this be characterized and shown more clearly?

4. Are there ever settings where we must achieve a particular retention group-wise while enforcing FPR per group to be less than some threshold? If so, how might one go about doing this?

5. For a group-conditional score, the efficient influence function for the \tau quantile in the nonparametric model can yield a one-step estimator. Can the authors comment on this and its feasibility?

6. Is there a retention vs FPR frontier per group? This could make the fairness-efficiency trade-off visually clear.

---

> ### Author Response · Authors · 2025-11-21
> **Response to Reviewer 4aeR**
>
> ### Answer for Question 1 (1/2)
>
> Thank you for raising these important points.
>
> **Data reuse and cross-fitting.**
> In MACI, the part related to “filter learning” is the optimization of the multi-LLM ensemble weights. As stated in line 1 of Algorithm 2 in the main text, we first split the available data into an optimization set $D_{\text{opt}}$ and a calibration set $D_{\text{cal}}$, and we only use $D_{\text{cal}}$ for conformal thresholding. Thus, nominal control is preserved in the original MACI pipeline.
>
> That said, we agree that cross-fitting is a very natural and appealing approach. We therefore conducted additional experiments applying a cross-conformal variant to the $D_{\text{opt}} / D_{\text{cal}}$ split, following the cross-conformal prediction procedure of Vovk et al. [9]:
> 1. Split the available data into $K$ folds.
> 2. For each fold $k$, use it as $D_{\text{cal}}^{(k)}$ and use the remaining $K-1$ folds as $D_{\text{opt}}^{(k)}$ to learn ensemble weights and perform conformal calibration.
> 3. Combine the resulting p-values from all $K$ weight-optimized ensemble models to perform filtering.
>
> We refer to this variant as **MACI-Cross**. The table below compares group-wise coverage and retention of MACI and MACI-Cross on the MedLFQA setting from Table 1 of the main text:
>
> |                     |         | **$\alpha = 0.2$** |              | **$\alpha = 0.1$** |              | **$\alpha = 0.05$** |              |
> | :-----------------: | :-----: | :----------------: | :----------: | :----------------: | :----------: | :-----------------: | :----------: |
> | **False-Claim Risk** | **Metric** | **MACI**           | **MACI-Cross** | **MACI**           | **MACI-Cross** | **MACI**            | **MACI-Cross** |
> | **Low**             | **Cov.** | 0.82 ± 0.03        | 0.81 ± **0.02** | 0.92 ± 0.03        | 0.92 ± **0.02** | 0.96 ± 0.03         | 0.96 ± **0.02** |
> |                     | **Ret.** | 0.71 ± 0.05        | **0.74** ± 0.02 | 0.46 ± 0.05        | **0.47** ± 0.02 | 0.33 ± 0.04         | **0.34** ± 0.03 |
> | **Medium**          | **Cov.** | 0.82 ± 0.03        | 0.83 ± **0.02** | 0.91 ± 0.03        | 0.91 ± **0.02** | 0.95 ± 0.02         | 0.96 ± **0.01** |
> |                     | **Ret.** | 0.70 ± 0.04        | 0.70 ± 0.01     | **0.51** ± 0.06    | 0.50 ± 0.03     | **0.38** ± 0.05     | 0.36 ± 0.03     |
> | **High**            | **Cov.** | 0.82 ± 0.03        | 0.82 ± **0.02** | 0.91 ± 0.02        | 0.91 ± **0.02** | 0.95 ± 0.02         | 0.95 ± **0.02** |
> |                     | **Ret.** | **0.59** ± 0.04    | 0.58 ± 0.02     | 0.38 ± 0.02        | 0.38 ± 0.01     | **0.25** ± 0.03     | 0.24 ± 0.01     |
>
> The results show that MACI and MACI-Cross have very similar coverage and retention across all risk groups and $\alpha$, while MACI-Cross slightly reduces variance. Unlike typical cross-conformal setups, MACI does not require retraining a classifier multiple times (e.g., repeated LLM fine-tuning); we only re-optimize ensemble weights over fixed verifier scores, so the additional cost is moderate. We will include MACI-Cross as the final version of MACI.
>
> **Covariate shift (distribution shift).**
> To account for covariate shift between calibration and test, we additionally construct explicit shift scenarios and implement the resampling-based density-ratio approach of Tibshirani et al. to estimate the density ratio with respect to the test distribution. We call this variant **MACI-DRE** and compare its group-wise coverage and retention against the original MACI. A detailed description of the methodology and results is provided in Section 5.3 and Appendix E of the revised manuscript.
>
> The table below shows that MACI-DRE recovers coverage and retention under covariate shift. Please refer to the methodology and experimental settings.
> | | | **$\alpha = 0.2$** | | **$\alpha = 0.1$** | | **$\alpha = 0.05$** | |
> | :---: | :---: | :---: | :---: | :---: | :---: | :---: | :---: |
> | **False-Claim Risk** | **Metric** | **MACI** | **MACI-DRE** | **MACI** | **MACI-DRE** | **MACI** | **MACI-DRE** |
> | **Low** | **Cov.** | 0.68 | 0.76 | 0.85 | 0.93 | 0.94 | 0.95 |
> | | **Ret.** | 0.83 | 0.72 | 0.57 | 0.35 | 0.34 | 0.29 |
> | **Medium** | **Cov.** | 0.65 | 0.84 | 0.82 | 0.94 | 0.87 | 0.96 |
> | | **Ret.** | 0.77 | 0.56 | 0.55 | 0.27 | 0.42 | 0.19 |
> | **High** | **Cov.** | 0.77 | 0.75 | 0.88 | 0.89 | 0.92 | 0.93 |
> | | **Ret.** | 0.62 | 0.67 | 0.39 | 0.38 | 0.27 | 0.28 |

---

> ### Author Response · Authors · 2025-11-21
> **Response to Reviewer 4aeR**
>
> ### Answer for Question 1 (2/2)
>
> As you noted in Weakness, MACI can experience greater variance in threshold estimation when some groups are small. MACI adopts a Mondrian-style group-conditional conformal scheme: for each pre-defined group $g \in G$, we calibrate a separate threshold using only the calibration examples in that group. This yields an exact finite-sample group-conditional guarantee under group-conditional exchangeability, but when the calibration sample size $n_g$ is small, the empirical $(1-\alpha)$-quantile can have high variance, leading to unstable thresholds and variability in both coverage and retention across groups.
>
> Following Gao et al. (2025) [8], we therefore construct a clustered variant, **MACI-Cluster**, that trades strict group-conditional guarantees for improved stability. On the MedLFQA dataset (the same base setting as in Table 1), we define fine-grained groups using the false-claim risk score and create a setting with many small groups (fewer than 50 data samples) by partitioning the data into 24 groups. We then:
>
> 1. Compute conformity-score histograms for each group.
> 2. Run $k$-means in this histogram space to obtain $K = 8$ clusters of groups.
> 3. Pool calibration scores within each cluster and estimate a single conformal threshold per cluster, which is shared by all groups assigned to that cluster.
>
> We compare MACI-Cluster with the original MACI in terms of empirical group-wise coverage and retention with 100 repeated runs.
>
> The details and results of this experiment are summarized in **Figure 5** and the **“Applying Group Clustering Method”** section of **Appendix E** in the revised manuscript. In short, MACI achieves group-wise coverage that, on average, is closer to the target level but with larger variance across groups, whereas MACI-Cluster has smaller variance and more stable thresholds, at the cost of relaxing strict group-conditional guarantees due to cluster-level pooling. As a consequence, MACI-Cluster is a good method for achieving high stability when the number of groups increases, without sacrificing strict group-conditional coverage.

---

> ### Author Response · Authors · 2025-11-21
> **Response to Reviewer 4aeR**
>
> ### Answer for Question 2, Weaknesses about Theorem 3 (1/2)
>
> We appreciate the reviewer for the detailed and insightful comments regarding the role of the constant appearing in the margin-type condition of Theorem 3. Since the concerns raised throughout the comments relate to the same underlying issue, we address them together below.
>
> **1. Clarifying the meaning of the constant and removing ambiguity.**
>
> The margin-type condition in Theorem 3 involves a constant denoted $C$, which may be confused with the notation for claim sets. In the revision, we therefore denote the population constant by $\mathfrak{C}$. More importantly, the constant in the condition
> $$
> \mathbb{P}\big(|p^*(P, c)-\tau| \leq \varepsilon\big) \leq \mathfrak{C} \varepsilon^\beta, \quad \forall \varepsilon>0,
> $$ is not tunable. It is a fixed structural characteristic of the true distribution of $D=(P,C,Y)$. The inequality must hold uniformly for all $\varepsilon>0$, which prevents the possibility of “making the assumption true” simply by selecting a larger $\mathfrak{C}$. Its value is entirely determined by how the oracle factuality score behaves in the neighborhood of the threshold $\tau$.
>
> We also note that in the earlier version of the Theorem 3, Assumption 1 included the condition $\mathbb{E}\big[(\hat{p}-p^{\ast})^{2}\big]<\infty$. This was a technical regularity condition that does not meaningfully restrict the analysis. If the mean squared error were unbounded, the final bound would be vacuous regardless of any other assumptions. For this reason, Assumption 1 has been removed from the statement of Theorem 1 in the revised manuscript.
>
>
> **2. Positioning the assumption within established statistical learning theory.**
>
> The condition used in Theorem 3 is a direct analogue of the classical margin assumptions that have become standard across modern statistical learning theory. These assumptions form the backbone of many fast-rate analyses in classification, plug-in estimation, and aggregation. Closely related formulations appear repeatedly throughout the literature, including
>
> [11] Optimal aggregation of classifiers (2004, Annals of Statistics),
>
> [12] Fast learning rates for plug-in classifiers (2007, Annals of Statistics),
>
> [13] Rates of Convergence for Nearest Neighbor Classification (2014, NeurIPS),
>
> [14] Regimes of No Gain in Multi-class Active Learning (2024, JMLR).
>
> Across these works, the constant analogous to $\mathfrak{C}$ is always treated as a population-level quantity rather than a tunable parameter. Its role is to characterize how the underlying distribution concentrates near the decision boundary, and our formulation plays the same role by describing how the oracle score $p^{\ast}$ behaves near $\tau$. The assumption therefore aligns exactly with the conventional margin conditions that are widely accepted and extensively used in statistical learning theory.

---

> ### Author Response · Authors · 2025-11-21
> **Response to Reviewer 4aeR**
>
> ### Answer for Question 2, Weaknesses about Theorem 3 (2/2)
>
> **3. Interpreting the condition when the oracle distribution is ambiguous near the threshold.**
>
> The reviewer noted that when the oracle factuality scores place substantial mass very close to the threshold $\tau$, the constant $\mathfrak{C}$ may become large. This interpretation is correct and reflects a property of the underlying distribution rather than an aspect of the assumption itself. The situation is analogous to classical quantile analysis, where limited separation around a target quantile naturally leads to weaker finite-sample guarantees. In our setting, if the oracle distribution assigns a high proportion of its mass to an $\varepsilon$-neighborhood of $\tau$, then the constants $(\mathfrak{C},\beta)$ must take larger values in order for the margin condition to hold.
>
> This behavior is determined entirely by the population distribution of $(P,C,Y)$. The constants in the margin condition describe how the oracle score behaves near the decision boundary and are not quantities that can be tuned or adjusted by the analyst. The condition therefore provides a clear description of how the intrinsic level of separation around $\tau$ influences the tightness of the efficiency bound. When the population distribution yields good separation around the threshold, the constants are favorable. When the oracle scores cluster near $\tau$, the constants reflect this increased ambiguity. The assumption does not impose density or smoothness requirements and simply formalizes this dependence on the distributional geometry of $p^{\ast}(P,c)$.
>
>
> **4. Why the constant cannot be tied to any observable or estimable quantity.**
>
> The reviewer asked whether $\mathfrak{C}$ could be related to an observable quantity, such as a plug-in estimator for a density at $\tau$. In our framework this is not possible. We impose no density or smoothness assumptions on $p^{\ast}(P,c)$, and the constant $\mathfrak{C}$ corresponds solely to how much mass $p^{\ast}$ assigns to an $\varepsilon$-neighborhood of $\tau$. This is not an estimable functional of the distribution, and the classical margin literature similarly treats the constants as population quantities rather than estimands.
>
> In practice, the only quantity that is both observable and relevant for the finite-sample bound is the estimation error: $\mathbb{E}\big[(\hat{p}-p^{\ast})^{2}\big]$. Once $(\mathfrak{C},\beta)$ are fixed by the population, the retention gap in Theorem 3 depends exclusively on how accurately $\hat{p}$ approximates $p^{\ast}$. When the oracle behaves favorably near $\tau$, even moderate accuracy in $\hat{p}$ results in minimal efficiency loss, whereas when oracle ambiguity is high, the difficulty is unavoidable for any method.

---

> ### Author Response · Authors · 2025-12-02
> **Response to Reviewer 4aeR**
>
> ### Answer for RCPS techniques and PAC bound
>
> Thank you for providing the insight that PAC guarantees can be derived due to the nature of the MACI algorithm. We are currently in the process of deriving the PAC bound you mentioned, and once the results are obtained and verified, we will add them to the revised version of the paper.

---

### Author Response · Authors · 2025-11-21
**Global Response**

We thank all four reviewers (`4aeR`, `eeqE`, `p4aF`, `fF7o`) for their careful and constructive feedback. This global response briefly summarizes the main strengths and concerns highlighted in the reviews, and explains how our revisions and additional experiments address them. Reviewer-specific clarifications and technical details are provided in the individual responses.

**Strengths highlighted by reviewers**

- [`4aeR`, `eeqE`, `p4aF`, `fF7o`] The paper presents a clear problem formulation and a well-motivated narrative for factuality filtering that jointly considers safety and utility.
- [`4aeR`, `eeqE`, `p4aF`, `fF7o`] The method demonstrates strong empirical performance with higher retention at target coverage and favorable time efficiency through a simple, one-pass calibration pipeline.
- [`eeqE`, `p4aF`] The proposed MACI method is novel, featuring a non-obvious yet natural multiplicative conformity score over claim-level factuality that aligns with the "no-false-claims" event.
- [`p4aF`, `fF7o`] Theorem 3 (Retention Ratio and MSE analysis) links oracle-score estimation error to true-claim preservation, providing a strong theoretical motivation for the multi-LLM ensemble design rather than treating it as a heuristic.
- [`eeqE`, `fF7o`] The paper is well-written with a clear and reasonable flow, ensuring that the empirical results and contributions are clearly communicated.

**Concerns and how we addressed them**

- **Small groups and group granularity** ([`4aeR`, `p4aF`]):
  We now explicitly discuss the trade-off between group granularity and sample size, and add a clustered variant (“MACI-cluster”) that calibrates on clusters of groups with similar score distributions. MACI keeps the sharpest group-conditional coverage, while MACI-cluster offers a more stable option when many small groups exist.

- **Assumptions 1–2 and the constant \(C\)** ([`4aeR`]):
  We clarify that these assumptions are a standard margin condition around the decision threshold. We explain how \(C\) and the margin exponent affect the tightness of the retention bound (especially for minority groups with low density near the boundary), and we explicitly describe this as a failure mode where clustering or pooling is recommended.

- **Covariate shift between calibration and test** ([`4aeR`, `p4aF`]):
  We add a “MACI under Covariate Shift” section with controlled shift scenarios and report group-conditional coverage and retention. We also introduce a lightweight density-ratio–weighted version of MACI (MACI-DRE), showing that simple importance weighting can substantially recover coverage under covariate shift.

- **Independence across claims from the same prompt** ([`fF7o`]):
  To test the conditional independence assumption, we introduce a joint-model variant (“MACI-Joint”) that models joint probabilities of claims per prompt. MACI and MACI-Joint exhibit similar coverage and only small differences in retention, indicating that MACI is not significantly affected by claim dependencies during the response generation phase.

- **Conformity-score design and robustness** ([`p4aF`]):
  We add an ablation on conformity-score variants, comparing our log-product score with log-sum and power-mean alternatives. All variants maintain coverage near the target, but our design consistently yields the best retention.

- **Factuality-score construction, ensemble details, and sampling-based baselines** ([ 'eeqE', `p4aF`, `fF7o`]):
  We clarify that we use multiple public LLMs as verifiers to produce base factuality scores, and combine them via an optimized weighted ensemble; we expand the description of how scores and weights are obtained and how ensemble diversity and efficiency are evaluated.

- **Related work on group-conditional CP, multicalibration, and fairness** ([`eeqE`, `fF7o`]):
  We expand the related-work section to cover multivalid conformal prediction, multicalibration, and recent group-conditional and multi-group UQ methods, and clarify that MACI targets high-retention group-conditional guarantees in a realistic factuality-filtering pipeline rather than conservative multivalid guarantees over many groups.

**We are doing our utmost to address the reviewers' questions and are proceeding with additional experiments and discussions deemed most critical. We will continue to update additional points throughout the remaining rebuttal period. We thank you once again for your interest and dedication to our paper.**

---

> ### Author Response · Authors · 2025-11-24
> **Global References**
>
> **The following references list all references used in the authors' response.**
>
> [1] Jung, Christopher, et al. "Batch Multivalid Conformal Prediction." International Conference on Learning Representations.
>
> [2] Liu, Terrance, and Steven Wu. "Multi-group Uncertainty Quantification for Long-form Text Generation." The 41st Conference on Uncertainty in Artificial Intelligence.
>
> [3] Hébert-Johnson, Ursula, et al. "Multicalibration: Calibration for the (computationally-identifiable) masses." International Conference on Machine Learning. PMLR, 2018.
>
> [4] Detommaso, Gianluca, et al. "Multicalibration for Confidence Scoring in LLMs." International Conference on Machine Learning. PMLR, 2024.
>
> [5] Ding, Tiffany, et al. "Class-conditional conformal prediction with many classes." Advances in neural information processing systems 36 (2023): 64555-64576.
>
> [6] Mohri, Christopher, and Tatsunori Hashimoto. "Language models with conformal factuality guarantees." arXiv preprint arXiv:2402.10978 (2024).
>
> [7] Cherian, John, Isaac Gibbs, and Emmanuel Candes. "Large language model validity via enhanced conformal prediction methods." Advances in Neural Information Processing Systems 37 (2024): 114812-114842.
>
> [8] Gao, Chenyin, Peter B. Gilbert, and Larry Han. "Bridging Fairness and Efficiency in Conformal Inference: A Surrogate-Assisted Group-Clustered Approach." Forty-second International Conference on Machine Learning.
>
> [9] Vovk, Vladimir, et al. "Cross-conformal predictive distributions." conformal and probabilistic prediction and applications. PMLR, 2018.
>
> [10] Tibshirani, Ryan J., et al. "Conformal prediction under covariate shift." Advances in neural information processing systems 32 (2019).
>
> [11] Tsybakov, Alexander B. "Optimal aggregation of classifiers in statistical learning." The Annals of Statistics 32(1) (2004): 135-166.
>
> [12] Jean-Yves Audibert and Alexandre B. Tsybakov. Fast learning rates for plug-in classifiers. The Annals of Statistics, 35(2):608–633, (2007).
>
> [13] Chaudhuri, Kamalika, and Sanjoy Dasgupta. "Rates of convergence for nearest neighbor classification." Advances in Neural Information Processing Systems 27 (2014).
>
> [14] Yuan, Gan, Yunfan Zhao, and Samory Kpotufe. "Regimes of no gain in multi-class active learning." Journal of Machine Learning Research 25.129 (2024): 1-31.
>
> [15] Manakul, Potsawee, Adian Liusie, and Mark Gales. "Selfcheckgpt: Zero-resource black-box hallucination detection for generative large language models." Proceedings of the 2023 conference on empirical methods in natural language processing. 2023.

---

### Author Response · Authors · 2025-12-01
**To New AC**

Dear New Area Chair,

We understand that due to the recent incident, you have been newly assigned to our submission and that all reviews and scores have been reverted to their pre-discussion state. Given that communication with the original reviewers is no longer possible, we are writing to provide a summary of the significant progress made during the rebuttal period to assist in your decision-making. Essentially, our rebuttal process was well-received, and the discussion had unanimously leaned towards acceptance. The concerns reflected in the currently visible scores were effectively resolved through our additional experiments and responses.

The status of each reviewer is as follows:

- Reviewer eeqE (Original: 4 $\rightarrow$ Raised: 6): This was the only borderline reject rating, primarily due to requests for experimental details and clarifications rather than logical flaws. Notably, the reviewer stated in their initial review that "I am open to improving my score". After we provided the requested clarifications and additional experiments (Appendix E), they explicitly confirmed: "Thank you for addressing my concerns, I will raise my score."

- Reviewer 4aeR (Original: 8): From the outset, this reviewer strongly supported acceptance, praising the problem formulation and practicality. Our responses to their technical questions regarding statistical stability further solidified their positive assessment.

- Reviewer p4aF (Original: 6): They raised valid questions about covariate shift in real-world deployments. We went beyond simple stress tests by proposing a Density-Ratio Estimation variant (MACI-DRE) and experimentally proving it recovers coverage. This technical depth effectively resolved their concerns.

- Reviewer fF7o (Original: 6): Despite a misunderstanding regarding the ensemble process, incorrectly assuming it required computationally expensive data re-generation, an interpretation that actually penalizes our method's efficiency, they still gave a positive score. We clarified that we ensemble scores on a fixed claim set, removing this perceived drawback.

Please note that the current scores indicate the status before the discussion. During the rebuttal, the reviewer eeqE expressed satisfaction with our responses, and **the general sentiment converged on acceptance**. We also provided detailed responses to reviewers 4aeR, p4aF, and fF7o, but unfortunately, we were **unable to receive their replies due to the incident**. We trust this summary of our rebuttal will assist your assessment.

Sincerely,

The Authors.

---

### Meta-Review · Area_Chair_Q5fj · 2025-12-19

**Summary:**

This paper is about using conformal inference to guarantee the factuality of claims generated by an LLM in a QA setting by filtering out generated claims that are deemed less likely to be factual. This follows notable work in the same area by improving the methodology to score the factuality of claims and filter them out, leading to higher retention and/or coverage than previous work.

Generally the reviewers agreed that this was a high quality work that brought both novel ideas, useful analysis, and sound experiments.

The main concerns from reviewers were around
- Incomplete literature review, especially around group-conditional conformal prediction, or methods used to improve the quality of generated LLM responses for retrieval or long-context settings.
- How the method might or might not handle covariate shift.
- How grouping is done and how the guarantees and results extend in this setting.
- Lack of some experimental details that made the work non-replicable
- Lack of relevant ablations over scoring methods/models
- Lack of runtime analysis

**Reviewer Concerns:**

- Lit review (partially addressed): The authors added relevant citations to some literature on group-conditional conformal which were clearly missed in the original, and used these ideas to improve their methods. However Reviewer p4aF also asked about RAG and long-context settings, which was not addressed. Paper [A] seems the most relevant here as it applies BCI to a RAG setting for both marginal and group-conditional conformal guarantees. The question about this method's compatibility with "very long contexts where claim segmentation and score independence may be more brittle" still seems relevant.

[A] Feng et al. "Response Quality Assessment for Retrieval-Augmented Generation via Conditional Conformal Factuality" SIGIR 2025

- Covariate shift (addressed): The original method is not designed to handle covariate shift, but the authors implemented a variation MACI-DRE adapting existing work to their setting to handle this.

- Grouping (addressed): More clarity was given that grouping was done using fixed labels for Mondrian CP, but the authors also adapted existing work that clusters groups, creating MACI-Cluster.

- Lacking details (addressed): More detail was given in Sec 5, and Reviewer eeqE responded that the concerns were answered.

- Lacking runtime (partially addressed): The authors briefly mentioned a single measurement of wall-clock time in the rebuttal. I do not think this robustly shows the tradeoffs between methods, and more careful analysis including a full description of what steps were included in the timing should be done before the authors claim that their method is more efficient than others.

**Reviewer Scores:**

My estimate of how the scores would have changed is as follows:

Reviewer 4aeR - 8 -> 8

Reviewer eeqE - 4 -> 6 This score raise was stated by the reviewer on Nov 25, slightly before the OpenReview bug was widely known.

Reviewer p4aF - 6 -> 6

Reviewer fF7o - 6 -> 6

---

### Decision · Program_Chairs · 2026-01-26

Accept (Poster)